# Gender differences in individual variation in academic grades fail to fit expected patterns for STEM

R.E. O'Dea [1,2], M. Lagisz[1], M.D. Jennions [2] & S. Nakagawa [1]

Fewer women than men pursue careers in science, technology, engineering and mathematics (STEM), despite girls outperforming boys at school in the relevant subjects. According to the 'variability hypothesis', this over-representation of males is driven by gender differences in variance; greater male variability leads to greater numbers of men who exceed the performance threshold. Here, we use recent meta-analytic advances to compare gender differences in academic grades from over 1.6 million students. In line with previous studies we find strong evidence for lower variation among girls than boys, and of higher average grades for girls. However, the gender differences in both mean and variance of grades are smaller in STEM than non-STEM subjects, suggesting that greater variability is insufficient to explain male over-representation in STEM. Simulations of these differences suggest the top 10% of a class contains equal numbers of girls and boys in STEM, but more girls in non-STEM subjects.

[1] Evolution and Ecology Research Centre, School of Biological and Environmental Sciences, University of New South Wales, Sydney 2052 NSW, Australia. [2] Research School of Biology, Australian National University, Canberra 2601 ACT, Australia. These authors contributed equally: R.E. O'Dea, M. Lagisz. Correspondence and requests for materials should be addressed to R.E.O'D. (email: rose.eleanor.o.dea@gmail.com) or to S.N. (email: s.nakagawa@unsw.edu.au)

A child entering school has endless answers to the question 'what do you want to be when you grow up?' By the end of school, these have narrowed to a set of career aspirations that are consistent with his or her self-concept (the way an individual perceives themselves, and believes they are perceived by others[1]). If the child is a girl, then she is likely to graduate with career aspirations with lower earning potential than a male classmate[2]. This phenomenon contributes to 'occupational segregation', and there are numerous incentives to reduce its prevalence. Schooling has a strong influence on the career aspirations of students[3], so addressing gender differences in the workforce requires that we understand how gender affects school achievement.

Self-concept is heavily influenced by school achievement[1,4], and high-performing students are more likely to pursue well-paid careers, such as science, technology, engineering and mathematics (STEM)-based jobs[5]. Girls tend to earn higher school grades than boys, including in STEM subjects[6], so why does this advantage not transfer into the workforce? The variability hypothesis, also called the greater male variability hypothesis, has been used to explain this apparent contradiction[7]—it is based on the tendency for males to show greater variability than females for psychological traits[8] (and for other traits across multiple species[9]), leading to relatively fewer females with exceptional ability[10]. However, the gender gap in employment within many highly paid occupations exceeds gender differences in variability (e.g. some math-intensive occupations employ far fewer women than the proportion of girls who score in the top 1% of maths tests[11]). Therefore, occupational segregation cannot be simply caused by fewer women having the requisite ability for high-status jobs.

Girls are susceptible to conforming to stereotypes (stereotype threat[12]) in the traditionally male-dominated fields of STEM, and girls who try to succeed in these fields are hindered by backlash effects[13]. STEM are high-paying fields that employ fewer women than men[14,15], and also require a high level of mathematical ability[16]. Evidence from standardised tests administered to children and adolescents indicates a greater gender difference in variation in performance in STEM subjects than other subjects[17–19], and an excess of males amongst the top-achieving students[20–22]. Therefore, a girl who performs well at school may notice that a greater proportion of the students who do better than her in mathematics and science classes are male, when compared to the proportion in other subjects. This, when combined with stereotype threat and the risk of backlash for behaving against gender stereotypes[13], could deter girls from pursuing a STEM-related career. Based on this hypothesis, and assuming equivalency of gender differences for standardised tests and class grades, we present an illustration of the predicted grade distributions for female and male students in Fig. 1.

Gender differences in variability have been tested using scores on standardised tests[19,23], but we are unaware of any study describing gender differences in the variability of teacher-assigned grades. While there are moderate-to-strong correlations (sensu[24]) between grades and test scores[25–28], there is also a stark gender difference. Girls tend to receive lower test scores relative to their school grades, whereas boys receive higher test scores relative to their school grades. There are multiple conjectures to explain this discrepancy in mean gender differences between tests and grades (e.g. on average, girls behave better, which gives them an advantage in grades, but they fare worse when tested on novel material that was not covered in class)[29]. Regardless of the source of these differences, teacher-assigned grades are likely to affect students' lives, and it is a reasonable conjecture that they have a greater impact on students' academic self-concept than standardised test scores[1]. Furthermore, grades are at least as good a predictor of success at university (measured by grade point average and graduation rate)[30,31]. Therefore, if gender differences in variability were impacting girls' decisions to pursue STEM, we would expect to see these differences reflected in school grades.

Here, we present a systematic meta-analysis on the effect of gender on variance in academic achievement using teacher-assigned grades. While grades are a more subjective measurement than test scores, we also include data from university students, whose grades are less affected by teachers' assessment of behaviour. While earlier meta-analyses have examined how mean academic achievement differs between the sexes[6,32,33], mean and variance differences should be examined together, as their magnitudes can be correlated (mean–variance relationship[34]). Fortunately, a recently published method allows for a meta-analytic comparison of variances that takes into account any mean–variance relationship[35].

Based on the variability hypothesis, we expected female grades to be less variable than those of males. To test this hypothesis, we extended a previous meta-analysis by Voyer and Voyer[6] on differences in the mean grades of students from ages 6 through to university. We used a more appropriate effect size to compare means, and another effect size to compare variances (Methods). We found that grades for female students were less variable than male grades. Then, focusing on school students (a relatively unbiased sample compared to university students), we found that: (1) the gender difference in variability has not changed noticeably over the last 80 years (1931–2013); (2) gender differences in grade variability are already present in childhood, and do not increase during adolescence; (3) finally, gender differences in grade variance were larger for STEM than non-STEM subjects, contrary to our expectations shown in Fig. 1.

## Results

**Description of dataset.** Our dataset contained 346 effects sizes extracted from 227 studies (Supplementary Data 1), representing 820,158 female and 826,629 male students. Fifty-two percent of the effect sizes were for 'global' grades (i.e. GPA), 26% were for STEM (mathematics and science), 19% for non-STEM (language, humanities, social science) and 3% for miscellaneous subjects. North American data dominated the dataset, with 70% of the effect sizes. Within the North American sample, 24% of studies were on a racially diverse cohort of students, 23% were on majority White/Caucasian students, 9% were on majority Black/African American students, 1% were on majority Hispanic/Latino students, and 43% of studies did not provide information on the racial composition of students. In total, 62% of the effect sizes came from school students (247,582 girls and 253,073 boys), and the remainder from university students. The original grades were awarded on a few different grading scales (Supplementary Figs. 1 and 2).

**Gender differences in variability.** Overall, girls had significantly higher grades than boys by 6.3% (natural logarithm of response ratio ($\ln RR_{overall}$(mean): 0.061, 95% confidence interval, CI: 0.052 to 0.070), with 10.8% less variation among girls than among boys (natural logarithm coefficient of variation ratio ($\ln CVR_{overall}$(variance): −0.114, CI: −0.133 to −0.095) (Supplementary Table 2; Fig. 2). The gender differences in mean grades were significantly larger at school than at university by 2.7% ($\ln RR_{school-uni\ diff}$: −0.028, CI: −0.044 to −0.011; Supplementary Table 3). The gender differences in variation were also larger at school than at university, but the difference of 4.2% was non-significant ($\ln CVR_{school-uni\ diff}$: 0.041, CI: 0.002 to 0.080; Supplementary Table 3). To test for moderating factors, we only used the school data in subsequent analyses. We excluded university students because there is self-selection among students in terms

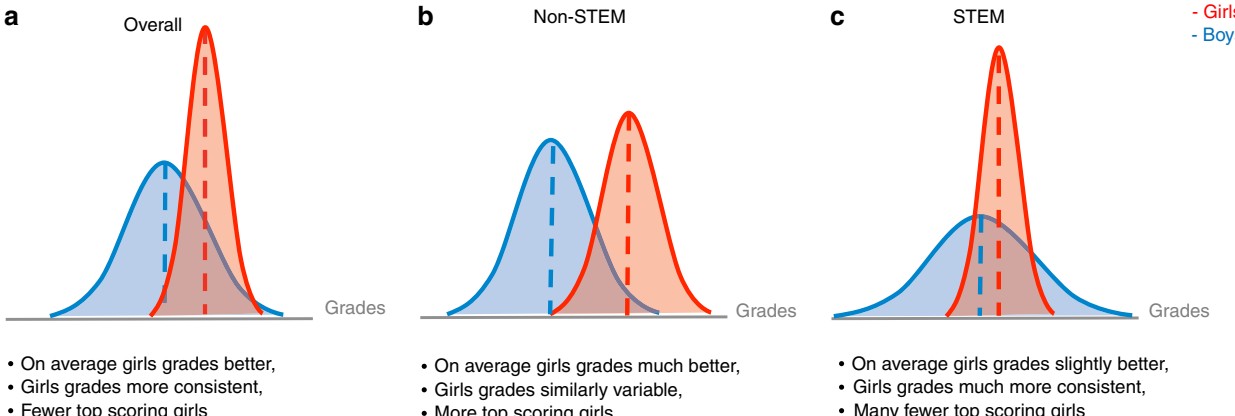

**Fig. 1** Predicted distributions of school grades of girls (red) and boys (blue). **a** The grade distribution overlaps represent the prediction that, when all grades are considered, girls on average earn higher grades and are less variable than boys, although there are more highly performing boys than girls at the upper end of the achievement distribution. **b** In non-STEM subjects, the difference in mean grades between girls and boys may be even more pronounced in favour of girls, which, coupled with similar variability, should result in many more highly performing girls than boys at the upper end of the achievement distribution. **c** In contrast, for STEM grades, we expected less difference between boys and girls mean grades and more grade variability for boys, resulting in boys dominating at both the top and bottom of the achievement distribution

of who applies for (and is then accepted at) a university. This selection process makes undergraduates and postgraduates unrepresentative of the general population. The results from analyses for the whole dataset and for the university subset are provided in Supplementary Tables 2–10, 12, and 15–25 (the university subset also had small sample sizes for STEM and non-STEM subjects, making results from moderator analyses sensitive to outlier studies).

**Moderating effects of study year and student age.** The higher mean and lower variability of girls' than boys' grades have not changed significantly over the past eight decades (Supplementary Table 4, Supplementary Fig. 8A: $\text{lnRR}_{\text{study year scaled (slope)}}$: 0.019, CI: −0.017 to 0.055; Supplementary Table 4; Supplementary Fig. 8D: $\text{lnCVR}_{\text{study year scaled (slope)}}$: −0.029, CI: −0.083 to 0.025). Within genders, variability in grades showed a non-significant trend towards decreasing over time, but significantly more so for girls than boys (Supplementary Table 5, Supplementary Fig. 8G: natural logarithms of the coefficient of variation $(\text{lnCV})_{\text{study year boys–girls (slope diff)}}$: 0.032, CI: 0.004 to 0.060). Student age did not affect the gap between girls and boys mean grades or the gender difference in grade variability (Supplementary Fig. 9, Supplementary Table 6). Within genders, variability in grades showed a non-significant tendency to decrease as students aged (Supplementary Table 7, Supplementary Fig. 9G: $\text{lnCV}_{\text{student age boys–girls (slope)}}$: 0.010, CI: −0.067 to 0.087), and to decrease faster for boys than girls (Supplementary Table 7, Supplementary Fig. 9G: $\text{lnCV}_{\text{student age boys−girls (slope diff)}}$: −0.035, CI: −0.062 to −0.007).

**Moderating effects of subject type: STEM versus non-STEM.** Girls' significant advantage of 7.8% in mean grades in non-STEM was more than double their 3.1% advantage in STEM. (Fig. 2a, Supplementary Table 8: non-STEM: $\text{lnRR}_{\text{non-STEM}}$: 0.075, CI: 0.049 to 0.102; STEM: $\text{lnRR}_{\text{STEM}}$: 0.031, CI: 0.011 to 0.051; the difference: $\text{lnRR}_{\text{non-STEM-STEM diff}}$: −0.044, CI: −0.065 to −0.024). Variation in grades among girls was significantly lower than that among boys in every subject type, but the sexes were more similar in STEM than non-STEM subjects (Fig. 2b, Supplementary Table 9; STEM: 7.6% less variable grades; $\text{lnCVR}_{\text{STEM}}$: −0.079, CI: −0.115 to −0.043; non-STEM: 13.3% less variable grades; $\text{lnCVR}_{\text{non-STEM}}$: −0.149, CI: −0.199 to −0.099; the difference: $\text{lnCVR}_{\text{non-STEM-STEM diff}}$: 0.070, CI: 0.028 to 0.111). The greater

gender similarity in variability in STEM was due to girls' grades being significantly more variable in STEM than non-STEM subjects (Fig. 2c, Supplementary Table 10, $\text{lnCV}_{\text{girls STEM–non-STEM diff}}$: −0.101, CI: −0.170 to −0.033). In contrast, the variability of boys' grades did not differ significantly between STEM and non-STEM subjects (Fig. 2c, Supplementary Table 10, $\text{lnCV}_{\text{boys STEM–non-STEM diff}}$: −0.030, CI: −0.102 to 0.042).

The small values of all meta-analytic estimates of gender differences in means and variances imply a large overlap in the grade distributions between the two sexes. The simulated distributions of girls' and boys' grades in Fig. 3 show the distributions of grades overlap more in STEM (94.2%) than non-STEM (88.2%) subjects. For example, within the top 10% of the distribution the gender ratio is even for STEM, and slightly female-skewed for non-STEM. Results of additional analyses are presented in Supplementary Tables 13–25.

## Discussion

Our overall result was consistent with elements of the variability hypothesis: female students' grades were less variable than those of male students, but in contrast to expectations, the greatest difference in variability occurred in non-STEM subjects. Average female grades were also higher than males, corroborating the findings of Voyer and Voyer[6] (Fig. 2). Gender differences in grade variability of school pupils was unaffected by their age, weakly affected by the year of study, and most strongly affected by whether or not the subject was STEM.

From grade one onward, we found that girls' grades were less variable than those of boys. Across the last 80 years, the variability in school grades has slightly decreased for both boys and girls (albeit slightly faster for girls). This decline might reflect increased student performance[36], or greater reluctance to fail students, i.e. grade inflation[37]. These scenarios assume that there is a ceiling effect on grades, whereby variance is reduced because weaker students are shifted upwards, whereas the highest performing students are bumped up against the 'ceiling' of the highest possible grade awarded on the grading scale. Although we do not see strong evidence for a ceiling effect in our dataset (Supplementary Fig. 5), below we discuss how the ceiling affect could underestimate the magnitude of gender differences in variability.

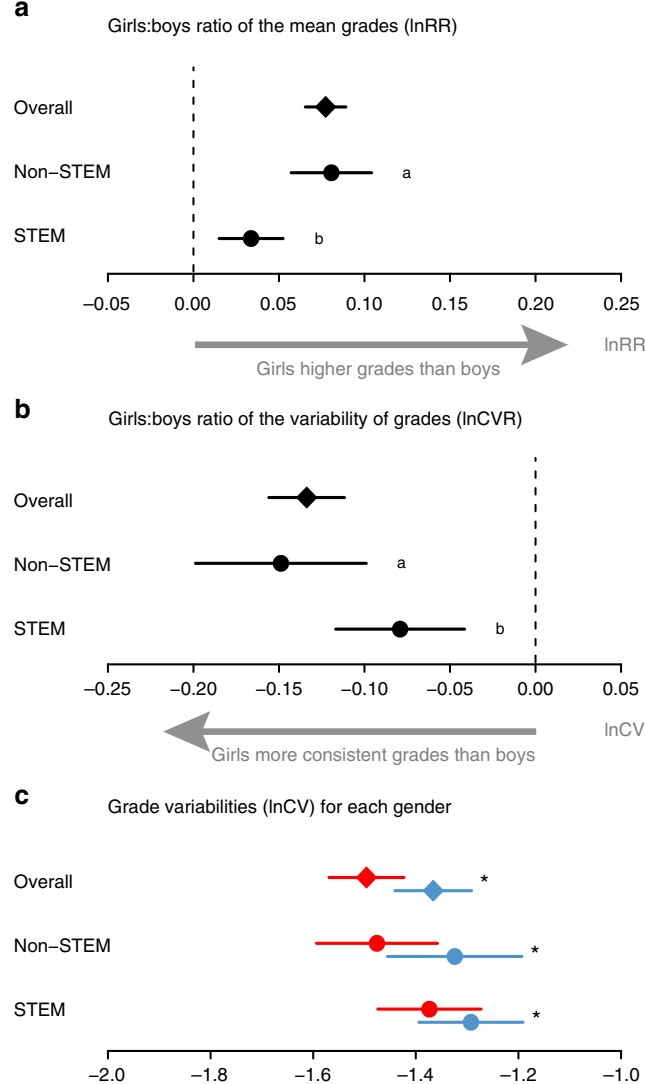

**Fig. 2** Main meta-analytic results. Results of analyses on **a** ratios of the grade means, **b** ratios of grade variabilities, and **c** coefficients of variations for girls (red) and boys (blue). Diamonds and circles represent meta-analytic estimates of mean effect sizes, and their 95% confidence intervals are drawn as whiskers. In **a**, natural logarithm of response ratio (lnRR) represents the average difference between girls' and boys' mean grades; positive values of lnRR indicate lower boys' mean grades. In **b**, natural logarithm coefficient of variation ratio (lnCVR) represents the average difference in grade variation between boys and girls; negative values of lnCVR indicate greater male variance. In **c**, natural logarithms of the coefficient of variation (lnCVs) are shown for boys and girls to illustrate grade variation by gender; more negative values of lnCV indicate less variation. Data and code for reproducing this figure are available at refs. [52,53]

Contrary to our expectations (Fig. 1), and those of many others[10], the gender difference in variability was smaller for STEM than non-STEM subjects (Fig. 2). When the small gender gap in grade variability is combined with the small gender difference in mean grades, it indicates that in STEM subjects, the distributions of girls' and boys' grades are more similar than in non-STEM subjects (Fig. 3). One possible explanation is that boys' are more affected by the ceiling affect in STEM than non-

STEM. For example, if a grading scale cannot distinguish between students in the top 1% or top 0.1%, and if there exists a male skew in the top 0.1% only in STEM but non in non-STEM, then gender differences in variance would be underestimated in STEM. Wai et al.[22] tried to get around this ceiling effect by analysing seventh-grade test scores explicitly designed to differentiate between exceptional students. They found a female:male ratio of 0.25 in the top 1% of students in STEM subjects, which is more imbalanced than our data suggests (Fig. 3c). While this finding is intriguing, it should be noted that STEM careers are not restricted to the exceptionally talented (although fields that subscribe to the belief that talent is important for success tend to employ fewer women[38]). Therefore, while our data does not preclude a gender gap among the exceptionally talented, it nevertheless indicates a practical similarity in girls' and boys' academic achievements, which are likely to provide an imperfect but valid measure of the ability to pursue STEM (Fig. 3).

Because students' grades impact their academic self-concept and predict their future educational attainment (e.g. refs. [1,5]), we might therefore predict roughly equal participation of men and women in STEM careers. However, the equivalence of girls' and boys' performance in STEM subjects in school does not translate into equivalent participation in STEM later in life. Is this because grades are not measuring the abilities required to succeed in STEM? Or does the relative advantage girls have over boys in non-STEM subjects at school lead them to rationally favour career choices with fewer competitors? We consider each of these questions in turn.

We analysed school grades, where girls show a well-established advantage over boys[25], whereas most previous tests of gender differences in variability have focussed on test scores[18,19,23]. To explore whether the smaller variability difference in STEM compared to non-STEM is confined to school grades, we performed a supplementary analysis of a large international dataset of standardised test scores of 15-year-olds (see Supplementary Note 2 for details). This supplementary analysis found gender differences in variance that were consistent across subjects; girls' test scores were more consistent than boys, with equivalent gender differences in non-STEM and STEM subjects (Supplementary Fig. 11). However, girls only showed a mean advantage in non-STEM. Therefore, it appears that the mean differences between test scores and grades are caused by shifts in the position of girls' and boys' distributions, rather than changes in the shape of distributions in STEM compared to non-STEM (girls' distributions of both grades and test scores are narrower than boys' distributions, but the difference is not more pronounced in STEM). If girls perceive they have fewer competitors in non-STEM subjects because, on average, fewer boys perform better than girls, this might lead to a preference for non-STEM over STEM careers[39,40].

Gender differences in expectations of success can arise due to backlash effects against individuals who defy the stereotype of their gender, and/or due to gender differences in 'abilities tilt' (having comparatively high ability in one discipline compared to another). Women in male-dominated pursuits, including STEM, face a paradox: if they conform to gender stereotypes, they might be perceived as less competent, but if they defy gender stereotypes and perform 'like a man', then their progress can be halted by 'backlash' from both men and women[13,41]. Furthermore, analyses of test scores have revealed that girls are more likely than boys to show an abilities tilt in the direction favouring non-STEM subjects (i.e. receive higher scores in non-STEM compared to STEM)[42]. Our data are consistent with girls showing an ability tilt in the direction of non-STEM subjects, although we cannot compare individual student grades (Supplementary Table 11). Intriguingly, there is evidence that balanced high-achieving

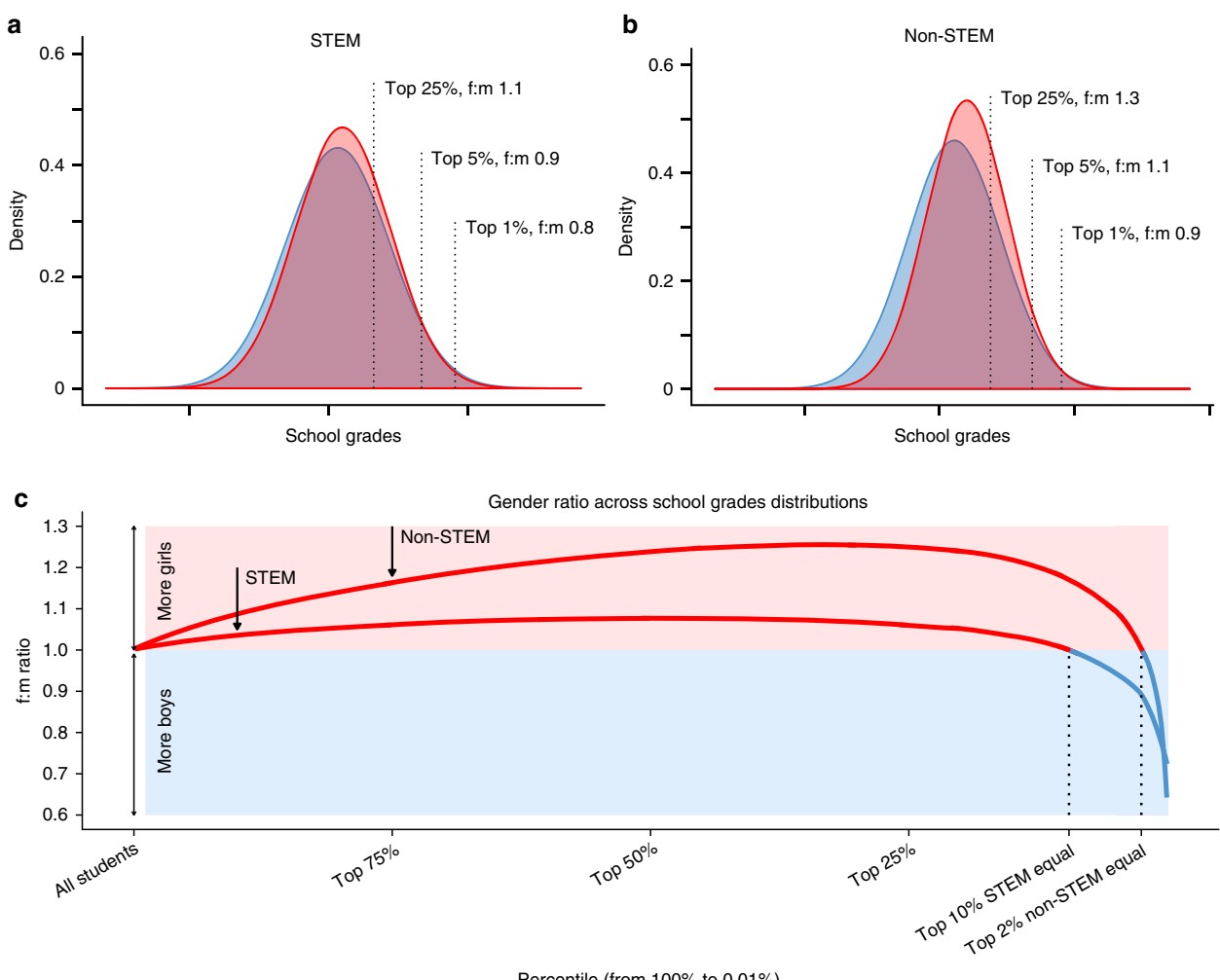

**Fig. 3** Inferred relative distributions of academic abilities of girls (red) and boys (blue). **a** STEM and **b** non-STEM school subjects, **c** the proportion of girls in each percentile. The relative mean and variance for each gender are based on the results of the meta-regression of school grades for school pupils, with subject as a moderator. In **a** and **b**, dashed vertical lines indicate cutoff points, above which 25%, 5% and 1% of top-scoring pupils can be found. The proportion of girls to boys across the distribution is shown in **c**, where values to the right on the x-axis correspond to the right tail of the achievement distributions. f:m values represent ratios of top-scoring girls to boys above each cutoff point (i.e, f:m > 1 = more females; f:m < 1 = more males). Data and code for reproducing this figure are available at refs. [52,53]

students—who possess the potential to succeed in disparate fields—prefer non-STEM careers[43], and that girls are more likely to be balanced than boys, at least among high achievers[44]. A female skew towards balanced abilities could be a manifestation of them showing lower levels of between-discipline variability (i.e. greater consistency across disciplines). Gender differences in between-discipline variability, rather than within-discipline variability, is an interesting avenue for future research.

A girl's answer to the question of 'what do you want to be when you grow up?' will be shaped by her own beliefs about gender, and the collective beliefs of the society she is raised in[45]. While our results support the variability hypotheses, we have shown that the magnitude of the gender gap in STEM grades is small, and only becomes male-skewed at the very top of the distribution (Fig. 3). Therefore, by the time a girl graduates, she is just as likely as a boy to have earned high enough grades to pursue a career in STEM. When she evaluates her options, however, the STEM path is trod by more male competitors than non-STEM, and presents additional internal and external threats due to her and societies' gendered beliefs (stereotype threat and backlash effects). To increase recruitment of girls into STEM, this path should be made more

attractive for them. A future study could estimate how male-skewed we would expect STEM careers to be based solely on gender differences in academic achievement, by quantifying the academic grades of current STEM employees. Our study focussed on gender differences in academic achievement, but understanding gender differences in any trait would be improved by simultaneously comparing gender differences in mean and in variability.

## Methods

**Literature search and study selection**. We performed a systematic literature search following guidelines from PRISMA (Preferred Reporting Items for Systematic Reviews and Meta-Analyses[46]). The PRISMA flow diagram depicting our search and screening process is shown in Fig. 4. We broadly followed the search protocol used by Voyer and Voyer[6]. We searched three databases for articles published between August 2011 and May 2015: ERIC, SCOPUS and ISI Web of Science. We did not use the PsychINFO or PsycARTICLES databases used by Voyer and Voyer[6], as they were malfunctioning at the time of our search. We searched for articles containing the term 'school grade/s', 'school achievement/s', 'school mark/s' or 'grade point average/s'. The exact search strings used for each database and additional details of the literature search are provided in Supplementary Methods. While there was no clear signal of publication bias in the school subset (Supplementary Tables 12, 25), a limitation of our literature search is that we did not actively search for unpublished studies or theses.

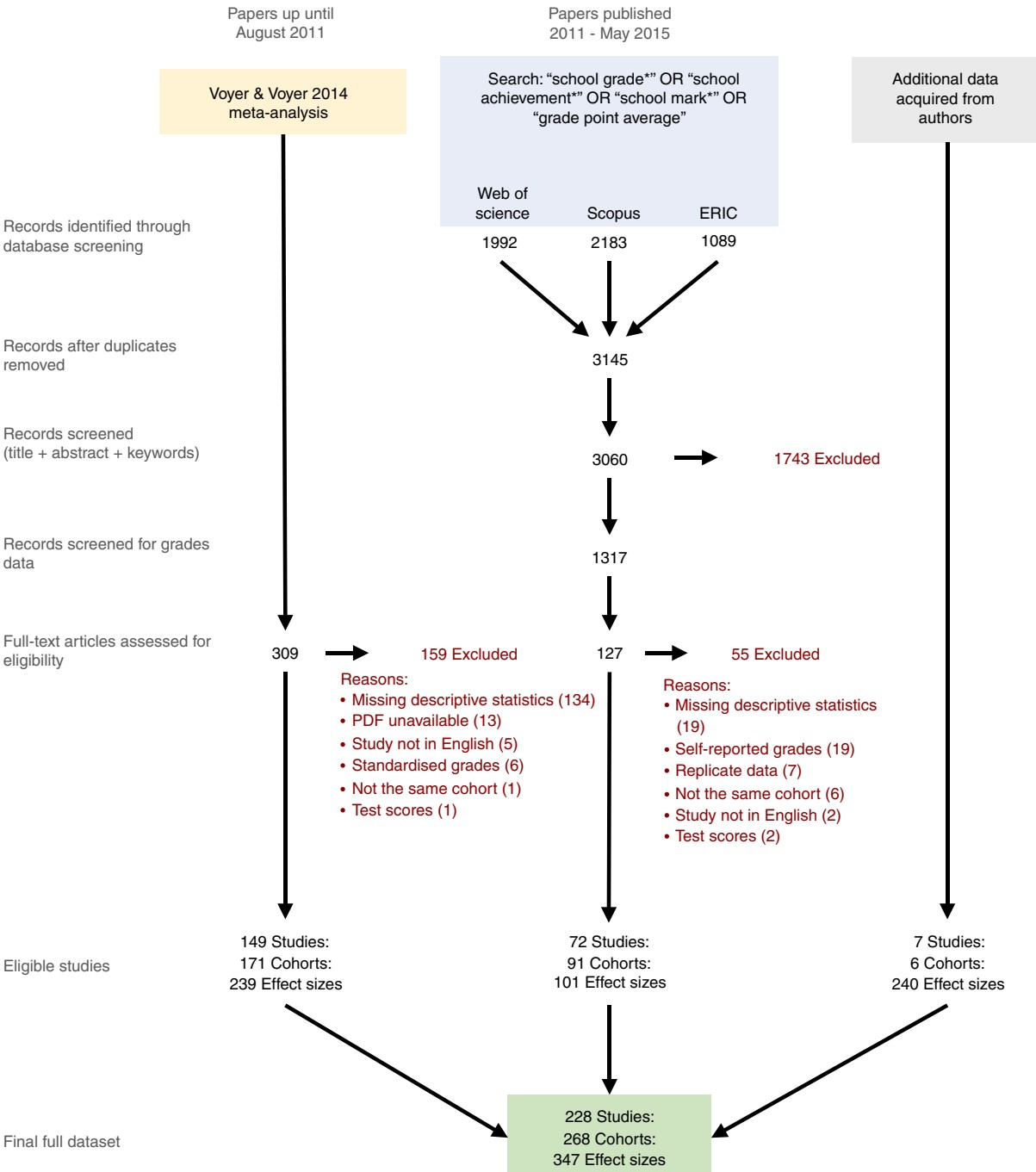

**Fig. 4** PRISMA diagram showing the process of locating studies included in this meta-analysis. The full list of included studies, and the list of studies excluded at the stage of full-text screening, are available in Supplementary Data 1 and 2

**Eligibility criteria**. To be included for data extraction at the full-text screening phase, studies needed to present teacher-assigned grades or global GPA (grade point average, i.e. grades averaged across many subjects) for a cohort containing both male and female students. The students could be from grade one and above. These criteria excluded kindergarten and single-sex studies, and self-reported grades or test data. Because of socio-cultural effects on gender differences, we required samples of students that took classes together; we therefore excluded online courses. We also excluded retrospective studies comparing adults that were not in the same study cohort. Where longitudinal data was reported, we included only the first year of data that met the inclusion criteria. In the case of studies that reported high school GPA for an undergraduate sample, we only included the university grades, if reported, and we deemed the high school grades ineligible. This is because the high school grades of groups of undergraduates do not come from the same cohort—they represent a subsample of students from disparate high schools, and only those students who performed well enough to attend university.

When we identified studies that reported data from the same large database, we only included the study with the largest sample size, and excluded the rest to avoid pseudo-replication. The list of excluded studies, with reasons for exclusion, is presented in Supplementary Data 2.

**Data extraction and coding**. From the original papers, we extracted the sample sizes, means, and standard deviations for male and female academic grades. For the studies used by Voyer and Voyer[6], we attempted to contact authors if any of these data were missing. All contacted authors were also asked to provide any additional data (published or unpublished) they might have available. If we received no response after 1 month, we sent a follow-up email. Only unstandardised grade data was collected. When presented data was standardised, we contacted authors to request the corresponding unstandardised values. For the studies published after August 2011, we only contacted authors if variance data was missing. In total, data from authors was acquired for 15 studies, including two unpublished studies.

**Moderator variables**. In addition to the descriptive statistics for grades of males and females, we extracted a number of moderator variables, all of which are presented in Supplementary Table 1. We generally followed the variables used by Voyer and Voyer[6] (e.g. racial composition), as well as recording additional information (e.g. age of students). An analysis of the moderating effect of racial composition on the gender gap in school grades is presented in the Supplementary Note 1 and Supplementary Tables 1, 3. Continuous moderators were scaled and centred (resulting in mean of 0, and standard deviation of 1) prior to the analyses. We used multiple imputations to fill in missing values of study year and students' mean age (details in Supplementary Methods).

**Effect sizes**. Using standardised effect sizes allowed us to combine original data collected on different scales (grades were recorded on different scales among included studies). To test for differences in mean grades between genders, we used the natural logarithm response ratio (hereafter referred to as lnRR), and its corresponding sampling error variance $s^2_{\text{lnRR}}$[47].

$$\text{lnRR} = \ln\left(\frac{\bar{x}_f}{\bar{x}_m}\right), \tag{1}$$

$$s^2_{\text{lnRR}} = \frac{s^2_m}{n_m \bar{x}^2_m} + \frac{s^2_f}{n_f \bar{x}^2_f}, \tag{2}$$

where:

$\bar{x}_f$ and $\bar{x}_m$ = the mean grade of female and male students, respectively,
$s^2_m$ and $s^2_f$ = the variance in grades of female and male students, respectively,
$n_m$ and $n_f$ = the number of male and female students in each sample, respectively.

Positive values of lnRR imply greater mean grades for girls.

We extended the literature search in Voyer and Voyer[6] by 5 years, and our analysis of mean grades differed from theirs in two ways: (1) we included only studies where we could compare variances, and; (2) we used lnRR instead of the standardised mean difference in performance (SMD or Hedges $g$[24]; see Supplementary Equations 1–4). We chose to use lnRR because, unlike SMD, it is unaffected by differences in variance (standard deviation) between groups. However, for comparison with Voyer's[6] results, we have repeated the lnRR analyses using SMD as the effect size. The results for both lnRR and SMD analyses—which are very similar to each other—are presented in the Supplementary Figure 4, and Supplementary Tables 2–4, 6, 8, 12, 13, 16, 19, 22, 25.

To assess differences in variance of grades of boys and girls, we used the natural logarithm coefficient of variation ratio (lnCVR) and its associated sampling error variance $s^2_{\text{lnCVR}}$[35].

$$\text{lnCVR} = \ln\left(\frac{CV_f}{CV_m}\right) + \frac{1}{2(n_f - 1)} + \frac{1}{2(n_m - 1)}, \tag{3}$$

$$\begin{aligned}
s^2_{\text{lnCVR}} = &\frac{s^2_m}{n_m \bar{x}^2_m} + \frac{1}{2(n_m-1)} - 2\rho_{\ln \bar{x}_m, \ln s_m} \\
&\times \sqrt{\frac{s^2_m}{n_m \bar{x}^2_m}\frac{1}{2(n_m-1)}} + \frac{s^2_f}{n_f \bar{x}^2_f} + \frac{1}{2(n_f-1)}, \\
&-2\rho_{\ln \bar{x}_f, \ln s_f} \sqrt{\frac{s^2_f}{n_f \bar{x}^2_f}\frac{1}{2(n_f-1)}}
\end{aligned} \tag{4}$$

where:

$CV_f$ and $CV_m$ = the coefficient of variation for males and females $\left(\frac{s}{\bar{x}}\right)$.
$\rho_{\ln \bar{x}_C, \ln s_C}$ and $\rho_{\ln \bar{x}_E, \ln s_E}$ = the correlations between the logged means and standard deviations of the male and female students, respectively.

All other notation is described above. Positive values of lnCVR imply greater variance in girls' grades relative to boys' grades. By dividing the female and male standard deviations by their respective means, we controlled for the effect of a proportional relationship (the mean–variance relationship) between the standard deviation and the mean. To test how the variance in grades has changed over time, we also computed the natural logarithm of the coefficient of variation (lnCV) for boys and girls separately, and its associated sampling error variance[35]:

$$\text{lnCV} = \ln\left(\frac{s}{\bar{x}}\right) + \frac{1}{2(n-1)}, \tag{5}$$

$$s^2_{\text{lnCV}} = \frac{s^2}{n\bar{x}^2} + \frac{1}{2(n-1)} - 2\rho_{\ln \bar{x}, \ln s}\sqrt{\frac{s^2}{n\bar{x}^2}\frac{1}{2(n-1)}}. \tag{6}$$

All notation as described above. For the same mean, a more negative value of lnCV implies a smaller variance.

**Statistical analyses**. We performed our main analyses on lnCVR and lnRR, and their associated error terms, using the rma.mv function in the R (v.3.4.2) package metafor v.2.0-0[48]. One-third of effect sizes were not independent, because they

came from the same study and/or the same cohort of students. We therefore included cohort ID and comparison ID as random effects in each model (the levels of study ID overlapped too much with cohort ID to model both levels simultaneously; e.g. in the school data, 120 studies and 141 cohorts, respectively). We also modelled covariance between effect sizes, assuming that effect sizes from the same cohort had 0.5 correlations between grades in different subjects (recommended in ref.[49]) because sampling error variances among these effect sizes based on the same cohort are likely to be correlated. We added this covariance matrix as our sampling error variance matrix (V argument in the rma.mv function). In addition, to account for the two main types of non-independence in our data (hierarchical/nested and correlation/covariance structures), we used the robust function within the metafor package to generate fixed effects estimates and confidence intervals, based on robust variance estimation, from each rma.mv model. To test for the overall effect of gender on mean and variance in school grades, we constructed meta-analytical models with no fixed effects (i.e. meta-analytic model or intercept-only model). We tested whether the results were significantly different between school and university by including the 'school or university' categorical moderator in a meta-regression model on the whole dataset. We then ran separate meta-analytical models on the school and university data subsets to quantify respective heterogeneities (Supplementary Methods). To test whether the gender gap in school grades varied between subjects, we included subject type (STEM, non-STEM, Global, Other/NR) as a fixed effect in meta-regression analyses. To test whether the gender difference in school grades has changed over historical time, or with student age, we included either study year or average student age as a fixed effect. To test whether the variance of either males or females has changed over historical time, or with student age, we used lnCV as the response variable, and the fixed effects of sex and study year, or sex and age, and their interactions. Point estimates from all statistical models were considered statistically significant when their CI did not span zero.

**Robustness of results**. There is a possibility of a bias in our results due to over-reporting of positive findings in published studies, so we tested our data for publication bias using multilevel-model versions of funnel plots and Egger's regression[50,51]. We also performed alternative analyses of key components of our study to test whether our conclusions are robust. Overlaps of grade distributions were inferred using simulation methods. Details and results of these analyses are presented in Supplementary Methods and Supplementary Tables 15–18, 20–23.

## Data availability

All data, code, and models that were used to generate results text, figures, and tables in the main text and supplementary information are available to download from dedicated repositories on the Open Science Framework[52,53].

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

## Acknowledgements

We thank the following authors for kindly providing data used in analysis: Dr. Stephen Borde, Dr. Christy Byrd, Dr. Christina Davies, Professor Rollande Deslandes, Professor Jean-Marc Dewaele, Professor Noor Azina Binti Ismail, Dr. Marianne Johnson, Dr. Amy Lutz, Dr. Amy Sibulkin, Dr. Helena Smrtnik Vitulić and Dr. Daniel Taylor. Sincere thanks to Dr. Khandis Blake, Dr. Daniel Noble, Dr. Joel Pick, Professor Cordelia Fine, for providing constructive comments that greatly improved the manuscript.

## Author contributions

S.N. and M.D.J. conceived the study, R.E.O. and M.L. collected data, R.E.O., M.L. and S. N. conducted analyses. All authors contributed to interpretation of the results and writing the manuscript.

## Additional information

**Competing interests:** The authors declare no competing interests.

