## [Peer Review File · Nature Communications]

Reviewers' comments:

Reviewer #1 (Remarks to the Author):

The authors extend Voyer & Voyer's (2014) meta-analysis by studying gender differences in variability in academic grades, in addition to mean differences (Voyer & Voyer analyzed differences in means, but not variability). Two key novel findings were that (a) boys' grades were more variable than girls' (b) but this gender difference was smaller in STEM than non-STEM subjects. Several large studies have investigated differences in variability and right tail performance for math test scores, but this paper is the first large study (to my knowledge) to do so for academic grades. The meta-analysis is technically sound, except for some minor questions and concerns, which I detail below.

However, my largest concern regards the study's more ambitious claims that these grades data show that "the gender bias in those pursuing careers in STEM cannot be attributed to more boys than girls showing exceptional talent" (abstract) and "the low rates of female recruitment into STEM careers is not due to differences in requisite ability" (page 12). Accepting these claims requires (1) accepting grades as a valid measure of STEM talent and ability (a questionable assumption) and (2) ignoring the overrepresentation of males in the right tail of math test performance found in other studies.

1. EQUATING GRADES WITH "TALENT, ABILITY": The authors explicitly state in the SI that they "assumed that school grades represent true academic abilities," but many researchers would strongly question this assumption. Grades can represent many constructs beside talent and ability such as classroom behavior, conscientiousness, politeness, perseverance, and sitting still and working quietly. Indeed, girls' superior social and behavioral skills have often been proposed as one explanation for girls' superior grades. For instance, one analysis of nationally representative data (Cornwell, Mustard, & Parys, 2012) found that "boys who perform equally as well as girls on reading, math and science tests are graded less favorably by their teachers." However, this gender gap disappeared once controlling for non-cognitive skills such as interpersonal skills and classroom engagement (link to study: <http://people.terry.uga.edu/cornwl/research/cmvp.genderdiffs.pdf>) . For these reasons, many researchers consider standardized test performance to be a better indicator of cognitive talent and ability than teacher-assigned grades.

My recommendation would be to describe the data more neutrally (as reflecting just teacher-assigned grades) and remove the broader claims about talent and ability.

2. MALE OVERREPRESENTATION IN HIGH MATH TEST PERFORMANCE: Boys outnumber girls by a ratio of about 2-4 to 1 in the top 5% or higher of mathematics test performance, even in recent years (e.g., Cimpian et al., 2016; Lakin, 2013; Wai et al., 2010). This basic finding – a male overrepresentation in the right tail of math test performance – is generally accepted among researchers, though its causes and implications are hotly debated. However, such findings are not discussed in this manuscript, though the authors would need to do so if they want to make broader claims about talent and ability (which again, I recommend they shouldn't here).

Cimpian, J. R., et al. (2016). Have gender gaps in math closed? Achievement, teacher perceptions, and learning behaviors across two ECLS-K cohorts. *AERA Open*, 2, 1-19.

Lakin, J.M. (2013). Sex differences in reasoning abilities: surprising evidence that male-female ratios in the tails of the quantitative reasoning distribution have increased. *Intelligence*, 41, 263–274.

Wai, J., et al. (2010). Sex differences in the right tail of cognitive abilities: a 30 year examination. *Intelligence*, 38, 412–423.

One study that is particularly relevant found that boys outnumbered girls in the right tail of math test performance (top 1%) as early as the fall of kindergarten (Cimpian et al., 2016). These findings counter the authors' claim that "gender differences in variability in academic performance among school-age children are poorly described...so it is unclear whether differences gradually emerge during school, or are already apparent earlier in childhood" (page 5). These studies also counter the abstract's claim that "previous tests of this explanation [i.e., a greater proportion of males than females are extraordinarily talented] are statistically inadequate."

Again, my basic recommendation would be to avoid making broader claims about ability and talent with these data. However, if the authors insist on discussing ability and talent, then these data on right tail differences in math test performance absolutely must be discussed. In other words, making claims about ability and talent with these data opens a Pandora's box of other research findings that would need to be discussed.

3. METHODS DETAILS: Below are minor, but still important, recommendations about describing study's methods and results.

(a) MAKE MATERIALS AVAILABLE: I strongly recommend that the authors upload their raw data and R analysis scripts as supplemental materials. Sharing such materials is good scientific practice and would also help answer questions I had about how specific analyses were conducted (e.g., how effect sizes were computed).

(b) EXPLAIN THE EFFECT SIZE METRICS: I recommend explaining the effect size metrics (lnRR and lnCVR) in greater detail because they will be unfamiliar to most readers. More specifically, I would present a specific concrete example of how descriptive statistics (M, SD, and n) were converted to these metrics and provide the equations used (e.g., $\lnRR = \ln(M_f/M_m)$) rather than just refer to a prior methods paper.

(c) MAKE THE EFFECT SIZES MORE INTERPRETABLE: One advantage of the lnRR and lnCVR metrics is that they can be interpreted readily in percentage terms (e.g., girls had 3.9% higher mean STEM grades but 7.8% less variability), as the authors do so on page 9. I recommend the authors do so earlier at the beginning of the results section on page 6. Interpreting the effect sizes in this way also makes clear that we are interpreting grades as ratio scale data (i.e., there is a true, not arbitrary, zero point that represents the minimum possible grades).

(d) ALSO PRESENT STANDARDIZED MEAN DIFFERENCES: I understand the authors' rationale for using lnRR over a standardized mean metric (e.g., Cohen's d), but using lnRR does make this study less directly comparable to the prior Voyer & Voyer (2014) meta-analysis. My recommendation would be to also present a standardized mean metric but in the supplemental materials.

(e) EXCLUSION OF KINDERGARTENERS: Why were kindergarteners excluded as noted on page 14 (lines 289-290)? They are school-age children, so I don't understand the rationale for excluding them.

(f) UNPUBLISHED STUDIES: One notable limitation of this meta-analysis and Voyer & Voyer's is the lack of search for unpublished studies through databases such as Google Scholar and ProQuest Theses & Dissertations. Obviously, the best solution to this issue would be to expand the literature search to more comprehensively search for unpublished studies. However, if the amount of work needed to do that is prohibitive, I would at the very least note the lack of unpublished studies as one important limitation of the literature search methods.

(g) NESTING OF EFFECT SIZES: The authors used the metafor R package to account for the nesting of effect sizes within studies (lines 344-348). However, the Hedges et al.'s (2010) robust variance estimation approach is generally considered to be a superior way to model such

dependencies (see <https://stackoverflow.com/questions/44811867/multilevel-meta-analysis-using-metafor> for discussion). I would recommend performing analyses using the `robumeta` R package which implements this method (Fisher & Tipton, 2015).

Fisher, Z., & Tipton, E. (2015, March 7). `robumeta`: An R-package for robust variance estimation in meta-analysis. Retrieved from <https://arxiv.org/pdf/1503.02220.pdf>

Hedges, L. V., Tipton, E., & Johnson, M. C. (2010). Robust variance estimation in meta-regression with dependent effect size estimates. *Research Synthesis Methods*, 1, 39-65. doi:10.1002/jrsm.5

Tipton, E. (2015). Small sample adjustments for robust variance estimation with meta-regression. *Psychological Methods*, 20, 375-393. doi:10.1037/met000011

Reviewer #2 (Remarks to the Author):

The paper addresses an interesting theoretical question - whether variability in student grades differs between STEM and non-stem subjects. In that regard it is novel, because most studies examine variability in performance on standardized tests (eg. NAEP, TIMSS, PISA), where there are small but non-trivial gender differences (favouring females for reading, males for some but not all samples of science and mathematics). But as the Voyer analysis concluded, females score considerably higher than males across all school subjects (including STEM)... i.e. the test-grade discrepancy. I think perhaps the authors have been too humble (or perhaps did not consider?) the novelty of their approach, but would be well to point this out to the reader at some point - by changing the outcome (DV) from achievement on tests to achievement on grades, you might get a completely different outcome. But it is also a potential limitation (that should be acknowledged) - the pattern of results may not replicate to performance on standardized tests of achievement.

Another novel aspect of this study is that they used an alternate statistical technique which combined mean gender differences with mean differences in variability (i.e. $\ln\text{CVR}$). Because this is a relatively new statistical technique, the authors should consider providing additional contextual information about the magnitude of this effect. $\ln\text{CVR}$ is a metric that few non-statisticians will immediately recognise, and so when you're making commentary about effect size (line 126, statistically significant but not quantifying the magnitude. line 179 The small values of all meta-analytic estimates of gender differences in means) you might want to provide contextual information about what size a medium or large effect size would look like. Otherwise its going to go over most reader's heads and the reader will be forced to rely on the author's judgement rather than deciding this for themselves. Figure 2C goes some way towards this goal, but some cutoffs or determinations made a priori would be helpful.

So putting aside the novel aspect of the study, there are some issues with the way literature is reviewed and the selection of references. The study is examining what the literature has termed the greater male variability hypothesis. The authors have consciously decided to reframe this as "lower female variability" and "females have less variance") which is a subjective but entirely legitimate approach to take (from a feminist perspective, why is it always framed has men have more), but I believe the authors should acknowledge (at some point, even if only briefly) the large body of research on this issue and mention the term at least once. It will bring greater attention (and hopefully citation) to the paper as that's the most common keyword search used. When I have to cite it, there's a particular reference I use that by Shields that might be helpful and I believe it would be compatible with the author's feminist perspective.

Shields, S. A. (1982). The variability hypothesis: The history of a biological model of sex differences in intelligence. *Signs*, 7(4), 769-797. doi: 10.2307/3173639

Some of the references are citing older research studies on mathematics, but completely neglected the issue of gender differences in science achievement (which I am sure the reader will agree, is germane to the issue of gender differences in STEM). For example you're citing a Hyde meta-analysis

Lindberg, S. M., Hyde, J. S., Petersen, J. L. & Linn, M. C. New Trends in Gender and Mathematics Performance: A Meta-Analysis. *Psychological Bulletin* 136, 1123–1135 (2010)

that analyses NAEP data, but didn't report the gender difference in science (which also differs across scientific discipline). Might I suggest (completely optional, as there are other studies available)

Reilly, D., Neumann, D. L., & Andrews, G. (2015). Sex differences in mathematics and science: A meta-analysis of National Assessment of Educational Progress assessments. *Journal of Educational Psychology*, 107(3), 645-662. doi: 10.1037/edu0000012

which also considers greater male variability and reports gender ratios at tails

or for cross-cultural analysis

Riegle-Crumb, C. (2005). The cross-national context of the gender gap in math and science. In L. V. Hedges & B. Schneider (Eds.), *The social organization of schooling* (pp. 227-243). New York, NY: Russell Sage Foundation.

Also there are typographical errors in the reference list that I noticed when following up the references

Line 373: Self-concept not Self Concept

Line 400: Published reference was X Chromosome not XChromosome

and frequent capitalization errors. I'm not sure what referencing style Nature Communications prefers, but they should at least be consistent

eg lines 379-386 compared to 388-390

These are minor issues, and only brought up to improve the quality of the final product. They do not significantly detract from the quality of the paper.

A final note: I always consider a good paper as being one that I would be likely to cite myself (once accepted for publication). This adds to the body of literature, and I very much look forward to seeing it in print (if not here, then certainly elsewhere).

Reviewer #3 (Remarks to the Author):

What are the major claims of the paper?

The major claims of the paper are that the higher proportion of boys relative to girls that are represented at the right-tail of the distribution in STEM grades are not sufficient to explain the higher proportion of males represented in STEM fields. In particular, the meta-analysis found that females were less variable than males across all grades but were more similar in mean and variance differences to males in STEM subjects than in non-STEM subjects. Therefore, the underrepresentation of females in STEM fields cannot solely be attributed to differences in the proportion of boys relative to girls performing at the highest levels in math and science.

Are the claims novel?

While meta-analyses of gender differences in verbal ability, math ability, and science ability have been conducted extensively, most meta-analyses have focused solely on mean differences. Therefore, the claims are novel enough given that the meta-analysis focuses on differences in variance as well.

Will the paper be of interest to others in the field?

Given that many STEM professionals perform in the top right-tail of the distribution of math ability, I believe a meta-analysis of variance differences in male/female distributions across STEM and non-STEM disciplines will be relevant and interesting to many in the field.

Will the paper influence thinking in the field?

The paper mostly confirms findings that have been demonstrated in prior literature suggesting that females have an advantage in grades over their male peers but show less variability in the overall distribution of scores. However, these findings reflect what I assume to be largely White/Caucasian samples, as African American and Latino/Hispanic males do not perform as highly as White or Asian males or as highly as their female counterparts. The findings may be bolstered by examination of race/ethnicity as a moderator and discussing the limitations of examining differences in male and female performance without considering the cultural context of race/ethnicity.

Are the claims convincing?

While the claims appear to convincingly convey the main points of the article, I do believe that a larger discussion regarding the limitations of using school grades or school marks as the indicator of ability is warranted. While grades may influence student self-concepts and task value and career aspirations, grades have also been heavily scrutinized by due to their overwhelming subjectivity. Multiple factors that are independent of actual ability influence the grades students obtain. Student behavior, for example, influences teacher assignment of grades. Girls tend to be less disruptive, more organized, and more studious, it has been suggested that girls' better behavior may be "one" factor contributing to their relatively higher grades across all subjects, including science and math. On standardized test scores in math and science, however, boys often outperform girls, and while grades may have a stronger influence on career aspirations, standardized test scores operate as gate-keepers for acceptance into competitive universities and many STEM majors. At the very least I believe the authors should address the limitations of examining grades as opposed to standardized test scores and discuss the pros and cons of each.

Are there other methods that would strengthen the paper further?

As suggested earlier, I believe that race/ethnicity should be examined as a moderator in these analyses.

Are the claims appropriately discussed in the context of previous literature?

As suggested earlier, the manuscript would benefit from discussion of the pros and cons of using school marks or grades as opposed to standardized test scores. It would also strengthen discussion of the findings to discuss how the overrepresentation of males in the right-tail distribution of non-STEM fields might impact female beliefs about their own abilities and career decisions. While women are not underrepresented in non-STEM fields they are underrepresented in positions that are believed to require innate intellectual ability or "brilliance" in both STEM and non-STEM fields (see the Science and Frontiers in Psychology articles by Cimpian, Leslie, & Meyers). Therefore, differences in the right-tail distribution may impact cultural values and beliefs about female ability across both fields, impacting women's career decisions in both fields. Greater discussion of these implications seems warranted.

Does the study seem sufficiently promising for a resubmission?

I recommend that the authors be invited to make revisions and resubmit the manuscript for publication.

**Reviewer 1, Comment 1**

*However, my largest concern regards the study's more ambitious claims that these*
*grades data show that "the gender bias in those pursuing careers in STEM cannot*
*be attributed to more boys than girls showing exceptional talent" (abstract) and*
*"the low rates of female recruitment into STEM careers is not due to differences in*
*requisite ability" (page 12). Accepting these claims requires (1) accepting grades as*
*a valid measure of STEM talent and ability (a questionable assumption) and (2)*
*ignoring the overrepresentation of males in the right tail of math test performance*
*found in other studies.*

**Response to Reviewer 1, Comment 1**

We have removed the sentence "the low rates of female recruitment into STEM
careers is not due to differences in requisite ability" from the discussion. We have
chosen to modify, rather than remove, the final line in the abstract, which now reads:
"the gender bias in those employed in standard careers in STEM is unlikely to be
driven by fewer girls than boys showing the requisite talent" (MT lines 22-22). The
modified sentence now allows that there may be more boys than girls who show
exceptional talent (as suggested by the right-tail studies the reviewer references), but
we do not believe this is one of the leading causes for the current magnitude of male
over-representation in STEM careers, which are not confined to those performing at
the elite level (see Response to Reviewer 3, Comment 3). Furthermore, we have
modified the manuscript to acknowledge and expand on your two points:

#####

(1) Are grades a valid measure of STEM talent and ability?

#####

In the revised manuscript, we touch on this question in the introduction before further
elaborating on it in the discussion.

*****

Introduction, MT Lines 86-89:

"...teacher-assigned grades affect student's lives, and they have a greater impact on
student's academic self-concept than standardized test scores (Möller et al. 2009).

Furthermore, grades are at least as good a predictor of success at university (measured
by GPA and graduation rate) (Betts & Morell 1999; Zhang et al. 2004). Therefore, if
gender differences in variability were impacting girls' decisions to pursue STEM, we
would expect to see these differences reflected in school grades."

*****

*****

Discussion, MT Lines 222-230:

**"Should equivalent grades translate into equivalent career choices?"**

"Because students' grades impact their academic self-concept and predict their future
educational attainment (e.g. ^{1.5}), we might therefore predict roughly equal
participation of men and women in STEM careers. However, the equivalence of girls'
and boys' performance in STEM subjects in school does not translate into equivalent

participation in STEM later in life. Is this because grades are not measuring the
abilities required to succeed in STEM? Or does the relative advantage girls have over
boys in non-STEM subjects at school lead them to rationally favour career choices
with fewer competitors? We consider each of these questions in turn.”

*****

We follow these questions with a discussion of our supplementary results for an
analysis of test scores (SI Lines 184-234) to show that we obtain qualitatively similar
results for gender differences in variance (we elaborate on these supplementary results
in our Response to Reviewer 1, Comment 2). Of course, it is possible that neither tests
nor grades represent a truly good measure of talent and ability in STEM, but given
they both set the bar for admission into higher education, and correlate with
completion of higher education degrees, they must be at least partly informative.

#####

(2) What about the over-representation of males in the right tail of test scores?

#####

This is a good point – in the revised introduction we now reference some of the
studies that have explicitly looked at the gender ratio amongst the very top achievers.

MT Lines 67-68: “...a male bias amongst the top-achieving students (Cimpian et al.
2016; Lakin 2013; Wai et al. 2010)”.

In the discussion we have added a paragraph that acknowledges the inability of our
 data to empirically quantify sex ratios at the top of the distribution (please see our
 “Response to Reviewer 3, Comment 3”). We can, however, simulate distributions
 based on our meta-analytic estimates, and estimate the sex ratio at different
 percentiles. We have now added this visualisation to Figure 3, which demonstrates
 how the very right tail of the grades distributions can become male-biased:

*****

Figures, MT Lines 607-608:

**“Figure 3”**

“Inferred relative distributions of academic abilities of girls (red) and boys (blue), for
 (A) STEM and (B) non-STEM school subjects, with (C) the proportion of girls in
 each percentile. The relative mean and variance for each gender are based on the

results of the meta-regression of school grades for school pupils, with subject as a
moderator. In (A) and (B), dashed vertical lines indicate cut-off points, above which
25%, 5% and 1% of top-scoring pupils can be found. The proportion of girls to boys
across the distribution is shown in (C), where values to the right on the x-axis
correspond to the right tail of the achievement distributions. f:m values represent
ratios of top-scoring girls to boys above each cut-off point (i.e. $f:m > 1$ = more
females; $f:m < 1$ = more males).”

*****

**Reviewer 1, Comment 2**

*EQUATING GRADES WITH “TALENT, ABILITY”:* The authors explicitly state
*in the SI that they “assumed that school grades represent true academic abilities,”*
*but many researchers would strongly question this assumption. Grades can*
*represent many constructs beside talent and ability such as classroom behavior,*
*conscientiousness, politeness, perseverance, and sitting still and working quietly.*
*Indeed, girls’ superior social and behavioral skills have often been proposed as one*
*explanation for girls’ superior grades. For instance, one analysis of nationally*
*representative data (Cornwell, Mustard, & Parys, 2012) found that “boys who*
*perform equally as well as girls on reading, math and science tests are graded less*
*favorably by their teachers.” However, this gender gap disappeared once*
*controlling for non-cognitive skills such as interpersonal skills and classroom*
*engagement (link to study:*
*<http://people.terry.uga.edu/cornwl/research/cmvp.genderdiffs.pdf>). For these*

*reasons, many researchers consider standardized test performance to be a better*
*indicator of cognitive talent and ability than teacher-assigned grades.*

...

*Again, my basic recommendation would be to avoid making broader claims about*
*ability and talent with these data. However, if the authors insist on discussing*
*ability and talent, then these data on right tail differences in math test performance*
*absolutely must be discussed. In other words, making claims about ability and*
*talent with these data opens a Pandora's box of other research findings that would*
*need to be discussed.*

**Response to Reviewer 1, Comment 2**

We agree that the assumption about grades representing true academic abilities is
overblown, and therefore we have modified the sentence in the SI to “assumed that
school grades represent student’s ability for future success in the academic discipline
(French et al. 2015)” (SI Lines 124-125). We also cite studies that show a correlation
between school grades and university performance in the introduction: “grades are at
least as good a predictor of success at university (measured by GPA and graduation
rate) (Betts & Morell 1999; Zhang et al. 2004)” (MT Lines 88-89).

We thank the reviewer for drawing attention to the interesting and relevant literature
on the links between behaviour, tests, and grades, and we have now alerted the reader

to these differences in the introduction, and cited the fascinating Cornwell 2012
paper, which the reviewer referred us to:

*****

MT Lines 77-86:

“Gender differences in variability have been tested using scores on standardized *tests*
(Reilly et al. 2015; Baye & Monseur 2016), but we are unaware of any study
describing gender differences in the variability of teacher-assigned *grades*. While
there are moderate-to-strong correlations (*sensu* Cohen, 1977) between grades and
test scores (Duckworth & M. Seligman 2005; McCandless et al. 1972; Borghans et al.
2016; Zwick & Green 2007) there is also a stark gender difference. Girls tend to
receive lower test scores relative to their school grades, whereas boys receive higher
test scores relative to their school grades. There are multiple conjectures to explain
this discrepancy in mean gender differences between tests and grades (e.g., on
average, girls behave better which gives them an advantage in grades, but they fare
worse when tested on novel material that was not covered in class) (Cornwell et al.
2013).”

*****

When we re-analyse the Cornwell data using our meta-analytic approach, we find that
the variance differences between grades and test scores are qualitatively similar, with
language subjects showing greater male variability than for maths or science (Figure
R1, below). The variability differences are actually larger in the test data compared to
the grades. What is most striking about the data is that behavioural scores show the
greatest gender differences in both mean and variance.

We include this analysis here for the reviewer's interest (see Figure R1, below), but
we have not added it to the manuscript (although we would be willing to incorporate
these results, if necessary). Instead, we have analysed a larger, international dataset of
test scores from the Project for International Student Assessment (PISA) – we have
elaborated on this analysis in our Response to Review 2 Comment 1 (please see
below). Both the Cornwell and PISA results show a similar pattern to our analysis of
grades data: overall boys are more variable (although this is marginally non-
significant in the test data from Cornwell), with the largest difference in variability
shown in non-STEM (reading or language) subjects. Therefore, it appears that gender
differences in variance are similar whether based on grades or standardised tests, but
this is less so for gender differences in means.

**Figure R1**

Re-analysis of data that appears in Cornwell et al. 2013, Table 1 (p.242), showing (A)

ratios of mean achievement for girls and boys, and (B) ratios of variability in

achievement for girls and boys. Diamonds and circles represent meta-analytic

estimates of mean effect sizes, and their 95% confidence intervals are drawn in

whiskers. In both panels, test scores are shown as purple diamonds, and grades and

shown as green circles. There is no test score for behaviour. In panel A, natural

logarithm of response ratio ($\ln RR$) represents the average difference between girls'

and boys' mean grades or test scores; positive values of *lnRR* indicate lower boys'
mean grades. In panel **B**, natural logarithm coefficient of variation ratio (*lnCVR*)
represents the average difference in grade or test score variation between boys and
girls; negative values of *lnCVR* indicate greater male variance.

**Reviewer 1, Comment 3**

*My recommendation would be to describe the data more neutrally (as reflecting just*
*teacher-assigned grades) and remove the broader claims about talent and ability.*

**Response to Reviewer 1, Comment 3**

In the revised manuscript we have emphasised the strengths and weaknesses of using
teacher-assigned grades, and we have shifted our tone from an emphasis on
talent/innate ability towards an emphasis on student's academic self-concept and
chance of future success (detailed in Response to Reviewer 1, Comment 1 and
Response to Reviewer 1, Comment 2, above).

**Reviewer 1, Comment 4**

**MALE OVERREPRESENTATION IN HIGH MATH TEST PERFORMANCE:**

*Boys outnumber girls by a ratio of about 2-4 to 1 in the top 5% or higher of*

*mathematics test performance, even in recent years (e.g., Cimpian et al., 2016;*

*Lakin, 2013; Wai et al., 2010). This basic finding – a male overrepresentation in the*

*right tail of math test performance – is generally accepted among researchers,*
*though its causes and implications are hotly debated. However, such findings are*
*not discussed in this manuscript, though the authors would need to do so if they*
*want to make broader claims about talent and ability (which again, I recommend*
*they shouldn't here).*

**Response to Reviewer 1, Comment 4**

We take the reviewer's point that school grades are not able to differentiate amongst
very high-achievers. The revisions we have made to address this point are detailed in
the above section under "Response to Reviewer 1, Comment 1": we have referenced
the male bias amongst top achievers in the introduction, and later in the discussion we
acknowledge that while girls are similar to boys in school grades, there may be gender
differences at the top of the distribution that we cannot identify. That said, in Figure
3C (see "Response to Reviewer 1, Comment 1") we explicitly show the expected
change in the female:male ratio as academic performance improves, based on
simulated distributions using our meta-analytic estimates of the mean and variance for
each gender.

The two main reasons for why we think our overall conclusions are valid are:

(1) Many factors are required to achieve in STEM, but we have not found

compelling evidence that one of these factors is exceptional talent/ability.

Even if girls' behaviours, such as conscientiousness, are accounting for their

high grades more than their high ability, these behaviours can have similar

benefits in later academic pursuits. We signal these points in the modified

introduction (MT lines 86-88: “teacher-assigned grades affect student’s lives,
and they have a greater impact on student’s academic self-concept than
standardized test scores (Möller et al. 2009). Furthermore, grades are at least
as good a predictor of success at university (measured by GPA and graduation
rate) (Betts & Morell 1999; Zhang et al. 2004”), and in the modified
discussion (MT line 215-216: “it should be noted that STEM careers are not
restricted to the exceptionally talented”).

(2) Even amongst the exceptionally talented, the gender imbalance is still smaller
than the gender imbalance in many STEM fields (Introduction, MT lines 54-
57: “However, the gender gap in employment within many highly paid
occupations exceeds gender differences in variability (e.g., some math-
intensive occupations employ far fewer women than the proportion of girls
who score in the top 1% of maths tests (Wang & Degol 2017))”).

**Reviewer 1, Comment 5**

*One study that is particularly relevant found that boys outnumbered girls in the*
*right tail of math test performance (top 1%) as early as the fall of kindergarten*
*(Cimpian et al., 2016). These findings counter the authors’ claim that “gender*
*differences in variability in academic performance among school-age children are*
*poorly described...so it is unclear whether differences gradually emerge during*
*school, or are already apparent earlier in childhood” (page 5). These studies also*
*counter the abstract’s claim that “previous tests of this explanation [i.e., a greater*
*proportion of males than females are extraordinarily talented] are statistically*
*inadequate.”*

**Response to Reviewer 1, Comment 5**

We thank the reviewer for citing the Cimpian 2016 study, which we have now cited in
the introduction (MT lines 67-68: "...a male bias amongst the top-achieving students
(Cimpian et al. 2016; Lakin 2013; Wai et al. 2010)". We have now removed both the
"statistically inadequate" sentence in the abstract, and the sentence in the introduction
about differences in school-age children (this part of the introduction has been
replaced with the paragraph about the differences between grades and test scores –
MT lines 77-91).

**Reviewer 1, Comment 6**

*MAKE MATERIALS AVAILABLE: I strongly recommend that the authors upload*
*their raw data and R analysis scripts as supplemental materials. Sharing such*
*materials is good scientific practice and would also help answer questions I had*
*about how specific analyses were conducted (e.g., how effect sizes were computed).*

**Response to Reviewer 1, Comment 6**

We agree with the reviewer. We had removed the link to our online data and code for
the double-blind review process, but we now realise we can anonymous, view-only

links, where you can download all data, code, and models presented in the
manuscript.

*****

MT Lines 418-420

**“Data and Code Availability”**

All data, code, and models are available from the following online repository:

https://osf.io/7btp3/?view_only=e69514d7ee544db8875a3822f9cbd505

*****

**Reviewer 1, Comment 7**

*EXPLAIN THE EFFECT SIZE METRICS: I recommend explaining the effect size*
*metrics (lnRR and lnCVR) in greater detail because they will be unfamiliar to most*
*readers. More specifically, I would present a specific concrete example of how*
*descriptive statistics (M, SD, and n) were converted to these metrics and provide the*
*equations used (e.g., $\lnRR = \ln(M_f/M_m)$) rather than just refer to a prior*
*methods paper.*

**Response to Reviewer 1, Comment 7**

We have now provided the equations for *lnRR*, *lnCVR*, and *lnCV* in the Materials and
methods section of the main text.

*lnRR:*

*****

MT Lines 345-350:

$$\ln RR = \ln \left(\frac{\bar{x}_f}{\bar{x}_m} \right)$$

$$s_{\ln RR}^2 = \frac{s_m^2}{n_m \bar{x}_m^2} + \frac{s_f^2}{n_f \bar{x}_f^2}$$

Where:

\bar{x}_f and \bar{x}_m = the mean grade of female and male students, respectively

s_m^2 and s_f^2 = the variance in grades of female and male students, respectively

n_m and n_f = the number of male and female students in each sample, respectively

*****

lnCVR:

*****

MT Lines 365-372:

$$\ln CVR = \ln \left(\frac{CV_f}{CV_m} \right) + \frac{1}{2(n_f - 1)} + \frac{1}{2(n_m - 1)}$$

$$s_{\ln CVR}^2 = \frac{s_m^2}{n_m \bar{x}_m^2} + \frac{1}{2(n_m - 1)} - 2\rho_{\ln \bar{x}_m, \ln s_m} \sqrt{\frac{s_m^2}{n_m \bar{x}_m^2} \frac{1}{2(n_m - 1)}} + \frac{s_f^2}{n_f \bar{x}_f^2} + \frac{1}{2(n_f - 1)} - 2\rho_{\ln \bar{x}_f, \ln s_f} \sqrt{\frac{s_f^2}{n_f \bar{x}_f^2} \frac{1}{2(n_f - 1)}}$$

CV_f and CV_m = the coefficient of variation for males and females $\left(\frac{s}{\bar{x}} \right)$

$\rho_{\ln \bar{x}_C, \ln s_C}$ and $\rho_{\ln \bar{x}_E, \ln s_E}$ = the correlations between the logged means and standard

deviations of the male and female students, respectively

All other notation is described above

*****

$\ln CV$:

*****

MT Lines 378-380:

$$\ln CV = \ln\left(\frac{s}{\bar{x}}\right) + \frac{1}{2(n-1)}$$
$$s_{\ln CV}^2 = \frac{s^2}{n\bar{x}^2} + \frac{1}{2(n-1)} - 2\rho_{\ln \bar{x}, \ln s} \sqrt{\frac{s^2}{n\bar{x}^2} \frac{1}{2(n-1)}}$$

All notation as described above.

*****

We have also provided the equations for Cohen’s *d* in the supplementary materials

(see “Response to Reviewer 1, Comment 9”).

Reviewer 1, Comment 8

***MAKE THE EFFECT SIZES MORE INTERPRETABLE:*** *One advantage of the*
*lnRR and lnCVR metrics is that they can be interpreted readily in percentage terms*
*(e.g., girls had 3.9% higher mean STEM grades but 7.8% less variability), as the*
*authors do so on page 9. I recommend the authors do so earlier at the beginning of*
*the results section on page 6. Interpreting the effect sizes in this way also makes*
*clear that we are interpreting grades as ratio scale data (i.e., there is a true, not*
*arbitrary, zero point that represents the minimum possible grades).*

**Response to Reviewer 1, Comment 8**

We agree with this point. We have now translated the meta-analytic means into
percentage increases, throughout the results section, in both the main text and the
supplement. For example:

*****

MT Lines 129-136:

**“Gender differences in variability”**

“Overall, girls had significantly higher grades than boys by 6.4% ($\ln RR_{\text{overall}}$ (mean):
0.062, 95% confidence interval, CI: 0.054 to 0.07), with 10.7% less variation among
girls than among boys ($\ln CVR_{\text{overall}}$ (variance): -0.113, CI: -0.132 to -0.095) (Table S3;
Figure 2). The gender differences in mean grades were significantly larger at school
than at university by 3% ($\ln RR_{\text{school-uni diff}}$: -0.031, CI: -0.047 to -0.015). The gender
differences in variation were also larger at school than at university, but the difference
of 4.8% was non-significant ($\ln CVR_{\text{school-uni diff}}$: 0.047, CI: 0.01 to 0.084; Table S4).”

*****

**Reviewer 1, Comment 9**

***ALSO PRESENT STANDARDIZED MEAN DIFFERENCES: I understand the***
***authors’ rationale for using lnRR over a standardized mean metric (e.g., Cohen’s***
***d), but using lnRR does make this study less directly comparable to the prior Voyer***

& Voyer (2014) meta-analysis. My recommendation would be to also present a
 standardized mean metric but in the supplemental materials.

**Response to Reviewer 1, Comment 9**

Following this recommendation, we have now provided results using Hedge’s *d*
 (standardised mean difference, SMD) (Hedges & Olkin, 1985) throughout the SI.

*****

Supplementary Methods and Results, SI Lines 154-169:

**“Direct comparison to Voyer & Voyer 2014: Standardised Mean Difference”**

“In addition to analysing gender differences in mean grades using *lnRR*, we also ran
 analyses using the standardised mean difference (*SMD*). This allows more direct
 comparison with the results of Voyer’s (2014) original meta-analysis, which used
 Cohen’s *d* (Cohen 1977) as the effect size. We have used a modified version of
 Cohen’s *d*, called Hedge’s *d* or *g* (Hedges & Olkin 1985), which has a bias correction
 for small sample sizes. We calculated *SMD* and its sampling variance, s_{SMD}^2 , as:

$$SMD = \frac{\bar{x}_f - \bar{x}_m}{s_{pooled}} J$$

$$J = 1 - \frac{3}{4(n_f + n_m - 2) - 1}$$

$$s_{pooled} = \sqrt{\frac{(n_f - 1)s_f^2 + (n_m - 1)s_m^2}{n_f + n_m - 2}}$$

$$s_{SMD}^2 = \frac{n_c + n_E}{n_c n_E} + \frac{SMD^2}{2(n_c + n_E)}$$

Where:

\bar{x}_f and \bar{x}_m = the mean grade of female and male students, respectively

s_m^2 and s_f^2 = the variance in grades of female and male students, respectively

n_m and n_f = the number of male and female students in each sample, respectively

Our results are similar to Voyer’s (2014) in both magnitude and significance, with the

exception that course material classified as “Other/NR” showed no significant mean

difference (our results: Table S17; Voyer’s results: Table 2, p.1189).”

*****

We find the results from SMD very similar to those from *lnRR*, which are now

mentioned in Materials and methods section of the main text:

*****

MT Lines 356-359:

“However, for comparison with Voyer’s⁶ results we have repeated the *lnRR* analyses

using *SMD* as the effect size. The results for both *lnRR* and *SMD* analyses – which are

very similar to each other – are presented in the SI.”

*****

**Reviewer 1, Comment 10**

*(e) EXCLUSION OF KINDERGARTENERS: Why were kindergarteners excluded*
*as noted on page 14 (lines 289-290)? They are school-age children, so I don't*
*understand the rationale for excluding them.*

**Response to Reviewer 1, Comment 10**

We followed the inclusion/exclusion criteria of Voyer's (2014) original meta-
analysis. They excluded kindergarten students, so we did too (p.1177 of Voyer &
Voyer 2014: "The specific criteria used in making inclusion decisions required
studies to have both male and female participants in Grade 1 (elementary school) or
later."). Voyer's paper does not provide rationale for excluding kindergarten, but we
think it might be because the meaning of kindergarten is ambiguous, both between
countries, and within countries; this term can refer to a variety of educational
institutions that may be separate from the primary school, and can be catering for
children across a wide age range. There is also a lack of consistent terminology (e.g.
preschool, pre-school, pre-primary, reception, transition), which makes searching and
screening for these studies extremely difficult.

**Reviewer 1, Comment 11**

*(f) UNPUBLISHED STUDIES: One notable limitation of this meta-analysis and*
*Voyer & Voyer's is the lack of search for unpublished studies through databases*
*such as Google Scholar and ProQuest Theses & Dissertations. Obviously, the best*
*solution to this issue would be to expand the literature search to more*

*comprehensively search for unpublished studies. However, if the amount of work*
*needed to do that is prohibitive, I would at the very least note the lack of*
*unpublished studies as one important limitation of the literature search methods.*

**Response to Reviewer 1, Comment 11**

This is true – we have now noted this limitation in the Methods section of our main
text.

*****

MT lines 297-299:

“While there was no clear signal of publication bias in the school subset in
supplementary analyses (see Table S13, Table S26), a limitation of our literature
search is that we did not actively search for unpublished studies or theses.”

*****

**Reviewer 1, Comment 12**

*(g) NESTING OF EFFECT SIZES: The authors used the metafor R package to*
*account for the nesting of effect sizes within studies (lines 344-348). However, the*
*Hedges et al.’s (2010) robust variance estimation approach is generally considered*
*to be a superior way to model such dependencies (see*
*[https://stackoverflow.com/questions/44811867/multilevel-meta-analysis-using-](https://stackoverflow.com/questions/44811867/multilevel-meta-analysis-using-metafor-for-discussion)*
*metafor for discussion). I would recommend performing analyses using the*
*robumeta R package which implements this method (Fisher & Tipton, 2015).*

**Response to Reviewer 1, Comment 12**

We thank the reviewer for pointing out this method and sending the related link.

Following this comment, we have re-analysed our data.

The main function for meta-analysis from the *robumeta* package – *robu* – cannot
completely deal with dependencies in our data. This is because we not only have
nesting of effect sizes within studies (i.e., hierarchical structure) but we also have
correlated effect sizes (e.g. STEM and non-STEM subjects; i.e., correlated structure).

We overlooked this correlated structure in our initial analysis, so in the revised
manuscript we have updated all our analyses to model sample variance as a
covariance matrix, assuming a 0.5 correlation between effect sizes from the same
cohort. The *robu* function is capable of modelling these two kinds of non-
independence, but not at the same time. We contacted the authors of *robumeta*, Dr
Fisher and Dr Tipton, who confirmed this shortcoming, at least for now, and advised
513 us to fit the hierarchical structure (as this is more important than the correlated
structure). We have run these models, and present them in an online repository:

*****

SI lines 88-92:

“We tested the robustness of our meta-analytical and meta-regression results using the
*robumeta R* package (Fisher & Tipton 2015) for robust variance estimation (RVE).

The results of these analyses are very similar to those obtained using the *robust*

function in the *metafor* R package, and they are available in this online
repository:
https://osf.io/ejqm4/?view_only=4c53bc91c2dd43879b0939827bf36fb3.”

****

We could incorporate these results directly into the main text or SI, if required.
For now, instead of presenting results from *robumeta*, we present results from models
that account for hierarchical structure, correlated structure, and robust variance
estimation using the *metafor* package. We have used the *robust* function, which
applies robust variation estimation to the model object from the *rma.mv* function. We
read about this *robust* function from the stackoverflow link mentioned in the
reviewer’s comment, for which we are very grateful. We note that this model also
uses robust variance estimation (we contacted the author of *metafor*, Dr Viechtbauer,
who confirmed this). We have therefore used the results from the original *rma.mv*
object to generate variance and heterogeneity estimates, and the results from the
*robust* output to generate fixed effects estimates. This method also has the advantage
of being able to provide residual variance estimates, and heterogeneity estimates,
which the *robu* function does not currently allow when modelling a hierarchical
structure (it provides these estimates for the default correlated structure method only).

We have modified the description of our analyses in the main text accordingly:

****

Materials and Methods, MT Lines 382-393:

**“Statistical Analyses”**

“We performed our main analyses on *lnCVR* and *lnRR*, and their associated error
terms, using the *rma.mv* function in the R (v.3.4.2) package *metafor* v.2.0-0
(Viechtbauer 2010). One-third of effect sizes were not independent, because they
came from the same study and / or the same cohort of students. We therefore included
cohort ID and comparison ID as random effects in each model (the levels of study ID
overlapped too much with cohort ID to model both levels simultaneously; e.g., in the
school data, 120 studies and 141 cohorts, respectively). We also modeled correlations
between effect sizes, assuming that effect sizes from the same cohort had 0.5
correlations between grades in different subjects (recommended in Noble et al. 2017)
and added this correlation matrix at the residual (comparison ID) level. In addition, to
account for the two main types of non-independence in our data (hierarchical/nested
and correlation structures), we used the *robust* function within the *metafor* package to
generate fixed effects estimates and confidence intervals, based on robust variance
estimation, from each *rma.mv* model.”

*****

**Reviewer 2, Comment 1**

**Reviewer 2:**

*The paper addresses an interesting theoretical question - whether variability in*
*student grades differs between STEM and non-stem subjects. In that regard it is*
*novel, because most studies examine variability in performance on standardized*
*tests (eg. NAEP, TIMSS, PISA), where there are small but non-trivial gender*

*differences (favouring females for reading, males for some but not all samples of*
*science and mathematics). But as the Voyer analysis concluded, females score*
*considerably higher than males across all school subjects (including STEM)... i.e.*
*the test-grade discrepancy. I think perhaps the authors have been too humble (or*
*perhaps did not consider?) the novelty of their approach, but would be well to point*
*this out to the reader at some point - by changing the outcome (DV) from*
*achievement on tests to achievement on grades, you might get a completely different*
*outcome. But it is also a potential limitation (that should be acknowledged) - the*
*pattern of results may not replicate to performance on standardized tests of*
*achievement.*

**Response to Reviewer 2, Comment 1**

We thank the reviewer for highlighting the novelty of testing the variability
hypotheses using school grades, rather than test scores. We have now highlighted this
point in the introduction:

*****

MT lines 77-79:

“Gender differences in variability have been tested using scores on standardized *tests*
(Reilly et al. 2015; Baye & Monseur 2016), but we are unaware of any study
describing gender differences in the variability of teacher-assigned *grades*.”

*****

Reviewer 1 also commented on the potential limitations of using grades rather than
tests. We have therefore discussed this distinction in both the introduction and
discussion – please see our comments above under “Response to Reviewer 1,
Comment 1”, and “Response to Reviewer 1, Comment 2” – while also highlighting
that grades are at least as valid a metric to examine gender differences as test scores.

In addition to these comments, we have also presented an additional analysis of test
data from the 2015 PISA. We discuss this analysis within one paragraph in the
discussion:

*****

MT lines 232-244:

“We analysed school grades, where girls show a well-established advantage over boys
(Duckworth & Seligman, 2006), whereas most previous tests of gender differences in
variability have focussed on test scores (Baye & Monseur 2016; Hedges & Nowell
1995; Reilly et al. 2015). To explore whether the smaller variability difference in
STEM compared to non-STEM is confined to school grades, we performed a
supplementary analysis of a large international dataset of standardised test scores of
15-year-olds (see the SI for details). This supplementary analysis found the same
pattern for gender differences in variance; girls’ test scores were more consistent than
boys, with the gender difference in variability being significantly greater in non-
STEM than STEM subjects (Figure S12). However, girls only showed a mean
advantage in non-STEM. Therefore, it appears that the mean differences between test
scores and grades is caused by shifts in the position of girls’ and boys’ distributions,
without changing the breadth of these distributions (girls’ distributions of both grades

and test-scores are narrower than boys’ distributions, but the difference is less
pronounced in STEM).”

*****

The details of this PISA analysis are covered in the Supplementary Methods and
Results, and a Figure and Table is also presented. All data, code and analyses used to
produce these results are available in this online repository:

https://osf.io/vu8h2/?view_only=b5131c883bd24b90919c4ea4944e93c5

*****

SI Lines 184-234

**“Analysis of test scores – 2015 PISA”**

“To explore whether the gender differences in variability across STEM and non-
STEM are broadly applicable to school achievement, and not just isolated to school
grades which tend to favour girls, we analysed data from the 2015 Programme for
International Student Assessment (referred to as PISA hereafter).

We downloaded data for reading, mathematics, and science subjects from the online
PISA Data Explorer (OECDa 2015), for students aged 15 years. We selected three
criteria: (1) “PISA Mathematics Scale: Overall Mathematics”; (2) “PISA Reading
Scale: Overall Reading”; and (3) “PISA Science Scale: Overall Science”. The
selected jurisdictions were “OECD”, “PARTNERS”, and “ADJUDICATED SUB-
REGION”, and each jurisdiction was split by the categorical variable of “Sex”. We
exported these results as three excel sheets, which provided the mean and standard
deviation of boys and girls in each of the jurisdictions for maths, reading, and science.

In order to obtain the sample sizes of males and females in each jurisdiction, we
downloaded the “Cognitive item data file” from PISA, in compressed SAS format
(OECDb 2015). We also downloaded the Codebooks for the main files in excel
format, to interpret the variable names in the cognitive data files. We imported the file
into *R* (v.3.4.3) using the *read_sas* function from the *haven* (v.1.1.1) package
(Wickham, H. & Miller 2018), and calculated the number of boys and girls who were
tested in each jurisdiction. Where a jurisdiction was not present in both datasets (data
explorer and cognitive item data file), it was removed.

Overall, the PISA dataset summarises test scores in maths, reading and science for
227,205 female and 226,293 male students. The students were tested from 63
jurisdictions. The minimum and maximum number of students in a jurisdiction was
3,371 and 23,141, respectively.

We calculated effect sizes for gender differences in mean and variance, using the
same metrics as our main meta-analysis (*lnRR*, *lnCVR*, and *lnCV*). We fitted meta-
analytic models to each effect size with the same approach as our main analysis
(using the *rma.mv* and *robust* functions from the *metafor* package (v. 2.0.0) in *R* (v.
3.4.3) (Viechtbauer 2010)). We accounted for non-independence arising from
multiple effect sizes from the same study by fitting the ID of the jurisdiction and a
comparison ID as random effects. We modeled a 0.5 correlation between test scores
from the same jurisdiction by including a correlation matrix at the residual
(comparison ID) level. To test for differences between STEM and non-STEM
subjects, we fit univariate meta-regression models by including subject as a fixed
effect, where subject was divided into STEM (maths and science) and non-STEM
(reading) categories.”

**“PISA Results”**

“Full results are shown in Table S15. Overall, girls sitting the 2015 PISA received
1.9% higher scores than boys ($\ln RR_{\text{overall}(\text{mean})}$ CI: 1.3% to 2.6%; Figure S12A), with
9.4% less variation among girls than among boys ($\ln CVR_{\text{overall}(\text{variance})}$; CI: 6.7% to 12%;
Figure S12B).

Girls’ small advantage in PISA tests scores was entirely driven by a 6.8% advantage
in non-STEM ($\ln RR_{\text{STEM}(\text{mean})}$ CI: 5.9% to 7.7%. In contrast, girls’ showed a non-
significant 0.4% *disadvantage* in STEM ($\ln RR_{\text{Non-STEM}(\text{mean})}$ CI: -1.0% to 0.2%; Figure
S12A).

While girls’ test scores were significantly more consistent across subjects, the
difference in non-STEM was 5.2% greater than the difference in STEM ($\ln CVR_{\text{STEM}}$
Non_STEM diff CI: 2.3% to 8.1%).

Both girls’ and boys’ scores were more variable in STEM than non-STEM subjects,
but the difference was smaller for girls, with a 5.5% increase compared to boys’
10.2% increase ($\ln CV_{\text{girls STEM non-STEM diff}}$ CI: 7.6% to 3.4%; $\ln CV_{\text{boys STEM non-STEM diff}}$ CI:
12.3% to 8%; Figure S12C).”

*****

*****

Supplementary Figures

SI Lines 321-333:

**“Figure S12”**

“Results of analyses on (A) ratios of the grade means, (B) ratios of grade variabilities,

and (C) coefficients of variations for girls (red) and boys (blue) from the 2015 PISA,

corresponding to Table S15. Diamonds and circles represent meta-analytic estimates

of mean effect sizes, and their 95% confidence intervals are drawn as whiskers. In

panel **A**, natural logarithm of response ratio (*lnRR*) represents the average difference
 between girls’ and boys’ test scores; positive values of *lnRR* indicate lower boys’ tests
 scores. In panel **B**, natural logarithm coefficient of variation ratio (*lnCVR*) represents
 the average difference in test score variation between boys and girls; negative values
 of *lnCVR* indicate greater male variance. In panel **C**, natural logarithms of the
 coefficient of variation (*lnCV*) are shown for boys and girls to illustrate grade
 variation by gender; more negative values on *lnCV* indicate less variation.”

*****
 *****
 *****

Supplementary tables

SI Lines 452-461

**“Table S15”**

“Estimated effect sizes and heterogeneity estimates for meta-analytic (intercept-only)
 models, and meta-regression models with subject (STEM or Non-STEM) as
 moderators, for PISA test scores. *lnRR* is a measure of mean difference between test
 scores for girls and boys. *lnCVR* is a measure of difference in variability between girls
 and boys. *lnCV* is a measure of test score variability relative to the mean grade for
 either girls or boys. I^2_{Total} represents proportion of variance not attributed to standard
 error. $I^2_{\text{Jurisdiction}}$ represents proportion of variance attributed to the Jurisdiction where
 students were tested. $I^2_{\text{comp_ID}}$ represents residuals against sampling error.”

Measure	Data	Fixed effects			Random effects			Heterogeneity		
		Mean	CI.lb	CI.ub		Sigma ₂	N levels	I^2_{Total}	$I^2_{\text{Jurisdiction}}$	$I^2_{\text{comp_ID}}$
lnRR	Intercept	0.019	0.013	0.025	Jurisdiction	0.000	63	98.5	0.0	98.5

	Subject: STEM	0.066	0.058	0.074	comp_ID	0.003		189		
	Subject: Non- STEM	-0.004	-0.010	0.002						
	non- STEM - STEM difference	-0.070	-0.074	-0.066						
lnCVR	Intercept	-0.098	-0.128	-0.069	Jurisdiction	0.003	63	98.1	18.0	80.1
	Subject: STEM	-0.132	-0.167	-0.097	comp_ID	0.015	189			
	Subject: Non- STEM	-0.082	-0.112	-0.051						
	non- STEM - STEM difference	0.050	0.023	0.078						
lnCV - girls	Intercept	-1.031	-1.115	-0.947	Jurisdiction	0.097	63	99.8	82.2	17.6
	Subject: STEM	-0.993	-1.077	-0.909	comp_ID	0.021	189			
	Subject: Non- STEM	-1.050	-1.135	-0.964						
	non- STEM - STEM difference	-0.057	-0.079	-0.034						
lnCV - boys	Intercept	-0.932	-1.015	-0.850	Jurisdiction	0.092	63	99.8	80.6	19.2
	Subject: STEM	-0.861	-0.946	-0.776	comp_ID	0.022	189			
	Subject: Non- STEM	-0.968	-1.051	-0.885						
	non- STEM - STEM difference	-0.107	-0.131	-0.083						

*****

**Reviewer 2, Comment 2**

*Another novel aspect of this study is that they used an alternate statistical technique*
*which combined mean gender differences with mean differences in variability (i.e.*
*lnCVR). Because this is a relatively new statistical technique, the authors should*
*consider providing additional contextual information about the magnitude of this*
*effect. lnCVR is a metric that few non-statisticians will immediately recognise, and*
*so when you're making commentary about effect size (line 126, statistically*
*significant but not quantifying the magnitude. line 179 The small values of all*
*meta-analytic estimates of gender differences in means) you might want to provide*
*contextual information about what size a medium or large effect size would look*
*like. Otherwise its going to go over most reader's heads and the reader will be*
*forced to rely on the author's judgement rather than deciding this for themselves.*
*Figure 2C goes some way towards this goal, but some cutoffs or determinations*
*made a priori would be helpful.*

We agree that this will be helpful for the reader, and Reviewer 1 made a similar point.
To indicate the magnitude of the effects we have now translated our estimates
differences into percentage differences, by taking the exponential of the natural
logarithm estimates (see “Response to Reviewer 1, Comment 8”), and we have also
provided the equations for the effect sizes within the manuscript (see “Response to
Reviewer 1, Comment 7”).

**Reviewer 2, Comment 3**

*So putting aside the novel aspect of the study, there are some issues with the way*
*literature is reviewed and the selection of references. The study is examining what*

*the literature has termed the greater male variability hypothesis. The authors have*
*consciously decided to reframe this as "lower female variability" and "females*
*have less variance") which is a subjective but entirely legitimate approach to take*
*(from a feminist perspective, why is it always framed as men have more), but I*
*believe the authors should acknowledge (at some point, even if only briefly) the*
*large body of research on this issue and mention the term at least once. It will bring*
*greater attention (and hopefully citation) to the paper as that's the most common*
*keyword search used. When I have to cite it, there's a particular reference I use that*
*by Shields that might be helpful and I believe it would be compatible with the*
*author's feminist perspective.*

*Shields, S. A. (1982). The variability hypothesis: The history of a biological model*
*of sex differences in intelligence. Signs, 7(4), 769-797. doi: 10.2307/3173639*

**Response to Reviewer 2, Comment 3**

This is a good point; we now say that the variability hypothesis is also called the
greater male variability when we first introduce it in the introduction (MT lines 50-51:
“The variability hypothesis, also called the greater male variability hypothesis”). We
have also added “greater male variability” to the list of keywords (MT lines 9-10).

**Reviewer 2, Comment 4**

*Some of the references are citing older research studies on mathematics, but*
*completely neglected the issue of gender differences in science achievement (which*
*I am sure the reader will agree, is germane to the issue of gender differences in*
*STEM). For example you're citing a Hyde meta-analysis*
*Lindberg, S. M., Hyde, J. S., Petersen, J. L. & Linn, M. C. New Trends in Gender*
*and Mathematics Performance: A Meta-Analysis. Psychological 457 Bulletin 136,*
*1123–1135 (2010)*
*that analyses NAEP data, but didn't report the gender difference in science (which*
*also differs across scientific discipline). Might I suggest (completely optional, as*
*there are other studies available)*
*Reilly, D., Neumann, D. L., & Andrews, G. (2015). Sex differences in mathematics*
*and science: A meta-analysis of National Assessment of Educational Progress*
*assessments. Journal of Educational Psychology, 107(3), 645-662. doi:*
*10.1037/edu0000012*
*which also considers greater male variability and reports gender ratios at tails or*
*for cross-cultural analysis*
*Riegle-Crumb, C. (2005). The cross-national context of the gender gap in math and*
*science. In L. V. Hedges & B. Schneider (Eds.), The social organization of*
*schooling (pp. 227-243). New York, NY: Russell Sage Foundation.*

**Response to Reviewer 2, Comment 4**

We thank the reviewer for these references. We have now cited the Reilly paper in the
introduction (MT lines 65-67: “. Evidence from standardised tests administered to
children and adolescents indicates a greater gender difference in variation in
performance in STEM subjects than other subjects (Feingold 1994; Hedges & Nowell
1995; Reilly et al. 2015)”) and in the discussion (MT lines 233-234: “most previous
tests of gender differences in variability have focussed on test scores (Baye &
Monseur 2016; Hedges & Nowell 1995; Reilly et al. 2015)”). The Riegle-Crumb
chapter was very interesting because it correlated girls’ attitude towards maths with
the level of gender stratification within society. We have placed this citation at the
head of the final paragraph of the discussion (MT lines 269-271: “A girl’s answer to
the question of “what do you want to be when you grow up?” will be shaped by her
own beliefs about gender, and the collective beliefs of the society she is raised in
(Riegle-Crumb et al. 2016).”).

**Reviewer 2, Comment 5**

*Also there are typographical errors in the reference list that I noticed when*
*following up the references*

*Line 373: Self-concept not Self Concept*

*Line 400: Published reference was X Chromosome not XChromosome*

*and frequent capitalization errors. I'm not sure what refercng style Nature*
*Communications prefers, but they should at least be consistent*

*eg lines 379-386 compared to 388-390*

*These are minor issues, and only brought up to improve the quality of the final*
*product. They do not significantly detract from the quality of the paper.*

**Response to Reviewer 2, Comment 5**

We have double-checked the reference formatting in our revised manuscript, and we
have also modified our list of references (some references have been added in the
course of revision, and some have been removed to reduce redundancy).

**Reviewer 3, Comment 1**

*The paper mostly confirms findings that have been demonstrated in prior literature*
*suggesting that females have an advantage in grades over their male peers but show*
*less variability in the overall distribution of scores. However, these findings reflect*
*what I assume to be largely White/Caucasian samples, as African American and*
*Latino/Hispanic males do not perform as highly as White or Asian males or as*
*highly as their female counterparts. The findings may be bolstered by examination*
*of race/ethnicity as a moderator and discussing the limitations of examining*
*differences in male and female performance without considering the cultural*
*context of race/ethnicity.*

**Response to Reviewer 3, Comment 1**

We agree that the interaction of racial composition and gender differences is an
interesting question, and it is important to contextualise our results with demographic
information, where it is available. Therefore, we have now included the racial
composition of the North American students in the first paragraph of our results
(“Description of dataset”).

*****

MT lines 122-125:

“Within the North American sample, 24% of studies were on a racially diverse cohort
of students, 23% were on majority White/Caucasian students, 9% were on majority
Black/African American students, 1% were on majority Hispanic/Latino students, and
43% of studies did not provide information on the racial composition of students”

*****

Unfortunately our dataset has limited potential to look for the effects of racial
composition, but we have included a supplementary analysis of gender differences
within the studies that report a majority racial composition of either black or white.
For the curious reader we also signal that this analysis exists in the main text
(Materials and Methods, MT lines 334-335: “An analysis of the moderating effect of
racial composition on the gender gap is presented in the SI.”). In brief, this analysis
does not show that gender differences are modified by racial composition.

In the Supplementary Methods and Results, we have included the following
paragraph:

*****

SI lines 171-182:

**“Do gender differences vary with racial composition?”**

“Following Voyer’s (2014) methods, for the North American samples we coded the
racial composition of each cohort if the original study reported a >75% racial majority
for the student population (for coding details, see Table S2). Twelve studies reported
a majority Black/African American racial composition (n = 15 effect sizes), and
eighteen studies reported a White/Caucasian majority (n = 36 effect sizes). No studies
reported an Asian American majority, and only two studies (n = 2 effect sizes)
reported a Latino/Hispanic majority, providing insufficient data for a comparison.
Therefore, to test for the effects of racial composition on gender differences in school
grades, we used the racial composition category of White/Black as a moderator
variable in univariate meta-regression models, We found no significant differences in
gender differences in either mean or variance (Table S14).”

*****

The full results are presented in the supplementary tables.

*****

SI lines 441-451:

“Table S14”

“Estimated effect sizes for univariate meta-regression models on gender differences in
 school grades among North Americans students, with race of students (>75% Black
 or White) as a moderator. *lnCVR* is a measure of grade consistency of girls and boys
 and *lnRR* and *SMD* are measures of mean grade difference between girls and boys. A
 positive contrast for *lnCVR* can be interpreted as White students having a smaller
 gender difference in grade variability than Black students, whereas a negative contrast
 for *lnRR* and *SMD* can be interpreted as White students having a smaller gender
 difference in mean grades than Black students. Estimates with confidence intervals
 (CI) not crossing zero are indicated in bold. *k* – number of effect sizes included.”

Measure	Data	Fixed effects			
		Mean	CI.lb	CI.ub	k
lnCVR	North American School subset				51
		Black	-0.175	-0.270	-0.080
		White	-0.150	-0.201	-0.100
		Black-White Contrast	0.025	-	0.132
lnRR	North American School subset				51
		Black	0.129	0.074	0.184
		White	0.079	0.055	0.103
		Black-White Contrast	-	-	0.010
SMD	North American School subset				51
		Black	0.492	0.384	0.600
		White	0.384	0.154	0.614
		Black-White Contrast	-	-	0.146

*****

**Reviewer 3, Comment 2**

*While the claims appear to convincingly convey the main points of the article, I do*
*believe that a larger discussion regarding the limitations of using school grades or*
*school marks as the indicator of ability is warranted. While grades may influence*
*student self-concepts and task value and career aspirations, grades have also been*
*heavily scrutinized by due to their overwhelming subjectivity. Multiple factors that*
*are independent of actual ability influence the grades students obtain. Student*
*behavior, for example, influences teacher assignment of grades. Girls tend to be*
*less disruptive, more organized, and more studious, it has been suggested that girls'*
*better behavior may be "one" factor contributing to their relatively higher grades*
*across all subjects, including science and math. On standardized test scores in math*
*and science, however, boys often outperform girls, and while grades may have a*
*stronger influence on career aspirations, standardized test scores operate as gate-*
*keepers for acceptance into competitive universities and many STEM majors. At the*
*very least I believe the authors should address the limitations of examining grades*
*as opposed to standardized test scores and discuss the pros and cons of each.*

**Response to Reviewer 3, Comment 2**

Both Reviewer 1 and Reviewer 2 made similar points about the distinction between
grades and tests. Please see our comments above: "Response to Reviewer 1,
Comment 1", "Response to Reviewer 1, Comment 2", and "Response to Reviewer 2,
Comment 1".

In brief, we have:

- • Added a paragraph in the introduction to highlight the differences between
grades and tests, cite some of the reasons for why these differences exist
(including behaviour), and also highlight the similarities between grades and
tests (they are correlated, and both predict later success at university)
- • Added a paragraph to the discussion to explore how one limitation on grades –
the ceiling effect – could have caused an underestimation of the variability
differences, by restricting the right tail of the distribution
- • Analysed standardised test data from the 2015 PISA, and find a qualitatively
similar result for gender differences in variability in STEM and non-STEM.
We present this analysis fully in the SI, and reference it in the discussion, in a
paragraph asking whether our result is generalizable beyond school grades.

**Reviewer 3, Comment 3**

*It would also strengthen discussion of the findings to discuss how the*
*overrepresentation of males in the right-tail distribution of non-STEM fields might*
*impact female beliefs about their own abilities and career decisions. While women*
*are not underrepresented in non-STEM fields they are underrepresented in*
*positions that are believed to require innate intellectual ability or “brilliance” in*
*both STEM and non-STEM fields (see the Science and Frontiers in Psychology*
*articles by Cimpian, Leslie, & Meyers). Therefore, differences in the right-tail*
*distribution may impact cultural values and beliefs about female ability across both*

*fields, impacting women's career decisions in both fields. Greater discussion of*
*these implications seems warranted.*

**Response to Reviewer 3, Comment 3**

This is an interesting hypothesis for why a male bias in the top 0.01% could
contribute to the wider male bias amongst the STEM workforce, despite the fact that
most of us are not particularly exceptional and did not score that highly.

In the revised discussion, we have touched on this point in a new paragraph, and cite
the 2015 Leslie et al. Science paper which found that disciplines that valued
giftedness more highly employed fewer women.

*****

MT lines 204-217:

“When the small gender gap in grade variability is combined with the small gender
difference in mean grades, it indicates that in STEM subjects the distributions of girls’
and boys’ grades are more similar than in non-STEM subjects (Figure 3). One
possible explanation is that boys’ are more affected by the ceiling affect in STEM
than non-STEM. For example, if a grading scale cannot distinguish between students
in the top 1% or top 0.1%, and if there exists a male bias in the top 0.1% only in
STEM but non in non-STEM, then gender differences in variance would be
underestimated in STEM. Wai et al.²² tried to get around this ceiling effect by
analysing seventh-grade tests scores explicitly designed to differentiate between
exceptional students. They found a female:male ratio of 0.25 in the top 1% of students
in STEM subjects, which is more imbalanced than our data suggests (Figure 3C).

While this finding is intriguing, it should be noted that STEM careers are not
restricted to the exceptionally talented (although fields that subscribe to the belief that
talent is important for success tend to employ fewer women (Leslie et al. 2015)).

*****

**References – Response letter**

Baye, A. & Monseur, C., 2016. Gender differences in variability and extreme scores
in an international context. *Large-scale Assessments in Education*, 4(1), p.541.

Betts, J.R. & Morell, D., 1999. The determinants of undergraduate grade point
average: the relative importance of family background, high school resources, and
peer group effects. *The Journal of human resources*, 34(2), pp.268–293.

Borghans, L. et al., 2016. What grades and achievement tests measure. *Proceedings of*
*the National Academy of Sciences*, 113(47), pp.13354–13359.

Cimpian, J.R. et al., 2016. Have gender gaps in math closed? achievement, teacher
perceptions, and learning behaviors across two ECLS-K cohorts. *AERA Open*,
2(4), p.233285841667361.

Cohen, J. *Statistical Power Analysis for the Behavioral Sciences (Revised Edition)*.
(Academic Press, 1977)

Duckworth, A.L. & Seligman, M., 2005. Self-discipline outdoes IQ in predicting
academic performance of adolescents. *Psychological science*, 16(12), pp.939–
944.

Duckworth, A.L. & Seligman, M.E.P., 2006. Self-discipline gives girls the edge:
Gender in self-discipline, grades, and achievement test scores. *Journal of*
*Educational Psychology*, 98(1), pp.198–208.

Feingold, A., 1994. Gender differences in variability in intellectual abilities: A cross-
cultural perspective. *Sex Roles*, 30(1-2), pp.81–92.

Fisher, Z. & Tipton, E., 2015. robumeta: An R-package for robust variance estimation
in meta-analysis. pp.1–16.

French, M.T. et al., 2015. What you do in high school matters: High School GPA,
educational attainment, and labor market earnings as a young adult. *Eastern*
*Economic Journal*, 41(3), pp.370–386.

Hedges, L.V. & Nowell, A., 1995. Sex differences in mental test scores, variability,
and numbers of high-scoring individuals. *Science*, 269(5220), pp.41–45.

Hedges, L.V. & Olkin, I., 1985. *Statistical Methods for Meta-Analysis*, San Diego.

Lakin, J.M., 2013. Sex differences in reasoning abilities: Surprising evidence that
male–female ratios in the tails of the quantitative reasoning distribution have
increased. *Intelligence*, 41(4), pp.263–274.

Leslie, S.-J. et al., 2015. Expectations of brilliance underlie gender distributions
across academic disciplines. *Science*, 347(6219), pp.262–265.

McCandless, B.R., Roberts, A. & Starnes, T., 1972. Teachers' marks, achievement
test scores, and aptitude relations with respect to social class, race, and sex.
*Journal of Educational Psychology*, 63(2), pp.153–159.

Möller, J. et al., 2009. A meta-analytic path analysis of the internal/ external frame of
reference model of academic achievement and academic self-concept. *Review of*
*Educational Research*, 79(3), pp.1129–1167.

Noble, D. W., Lagisz, M., O’Dea, R. E. & Nakagawa, S., 2017. Nonindependence and
sensitivity analyses in ecological and evolutionary meta-analyses. *Mol. Ecol.* 26,
pp.2410–2425.

OECDa. PISA Data Explorer. (2015). OECD Skills Surveys,

<http://pisadataexplorer.oecd.org/ide/idepisa/>, accessed on 26th February 2018.

OECDb. PISA 2015 Database. (2015). OECD Skills Surveys,

<http://www.oecd.org/pisa/data/2015database/>, accessed on 26th February

2018.

Reilly, D., Neumann, D.L. & Andrews, G., 2015. Sex differences in mathematics and

science achievement: A meta-analysis of National Assessment of Educational

Progress assessments. *Journal of Educational Psychology*, 107(3), pp.645–662.

Riegle-Crumb, C., King, B. & Moore, C., 2016. Do they stay or do they go? The

switching decisions of individuals who enter gender atypical college majors. *Sex*

*Roles*, 74(9), pp.436–449.

Viechtbauer, W., 2010. Conducting meta-analyses in R with the metafor package.

*Journal of Statistical Software*, 36(3), pp.1–48.

Voyer, D. & Voyer, S.D., 2014. Gender differences in scholastic achievement: a

meta-analysis. *Psychological Bulletin*, 140(4), pp.1174–1204.

Wai, J. et al., 2010. Sex differences in the right tail of cognitive abilities: A 30 year

examination. *Intelligence*, 38(4), pp.412–423.

Wang, M.T. & Degol, J.L., 2017. Gender gap in science, technology, engineering, and

mathematics (stem): current knowledge, implications for practice, policy, and

future directions. *Educational Psychology Review*, 29(1), pp.119–140.

Wickham, H. & Miller, E., 2018. haven: Import and Export 'SPSS',

- 'Stata' and "SAS" Files. R package version 1.1.1.
- Zhang, G. et al., 2004. Identifying factors influencing engineering student graduation:
A longitudinal and cross-institutional study. *Journal of Engineering Education*,
93(4), pp.313–320.
- Zwick, R. & Green, J.G., 2007. New Perspectives on the Correlation of Scholastic
Assessment Test Scores, High School Grades, and Socioeconomic Factors.
*Journal of Educational Measurement*, 44(1), pp.1–23.

Reviewers' comments:

Reviewer #1 (Remarks to the Author):

I commend the authors for their comprehensive response to my and other reviewers' earlier comments. This study continues to address an important research question that has been studied before with test scores, but not grades, making the findings novel and interesting. Figure 3 was a great addition that was very clear. I also especially appreciate that the authors made the data and analysis scripts available. With those materials, I (mostly) verified the empirical findings, with some minor exceptions that I detail later.

Many of my earlier comments and concerns have now been addressed (e.g., statistical models to address correlated effects, effect size computations). However, two major concerns remain: (1) the claims about talent and ability, and (2) the new PISA test score analysis raises more questions than it answers. The manuscript has been improved in the first regard, but it still makes claims that go beyond the scope of this project, especially in the abstract (which is what most readers will see). The second concern is new because the PISA test score analysis is new.

If these two major concerns are addressed, I would recommend publication, but I cannot do so in the current state. In addition, I detail smaller concerns and comments to help further improve the manuscript, but those are less essential to address.

(1) CLAIMS ABOUT TALENT AND ABILITY

The manuscript has been improved now that it explicitly acknowledges the test-grade discrepancy and potential reasons for it. The claims about talent and ability are now far more nuanced and appropriate given the complexity of the broader data. However, there are places still where the claims go beyond the data by inappropriately conflating grades with talent and ability.

THE ABSTRACT. My biggest concern is the last sentence of the abstract: "Given the similar distribution of STEM grades for girls and boys, the gender bias in those employed in standard careers in STEM is unlikely to be driven by fewer girls than boys showing the requisite talent." The manuscript later adds nuance to this claim by noting how student behavior can influence grades (e.g., lines 84-85). However, the abstract must be evaluated as a critical standalone paragraph, and in this case the claims go beyond the current focus and scope of this project.

MANUSCRIPT: In addition, there are places in the manuscript where grades are referred to "academically measured ability" (lines 217-220) and "academic ability" (lines 273-275). I applaud the authors for emphasizing how grades might shape self-concepts and predict later outcomes such as graduating college (those are compelling points). But still I'd argue referring to grades as "ability" is not appropriate.

My concerns echo Reviewer #3's earlier comment: "grades have also been heavily scrutinized by due to their overwhelming subjectivity. Multiple factors that are independent of actual ability influence the grades students obtain. Student behavior, for example, influences teacher assignment of grades. Girls tend to be less disruptive, more organized, and more studious."

MY RECOMMENDATION: My recommendation is simple: avoid making claims about talent and ability in the abstract. The study is sufficiently noteworthy without having to make such claims (and, in fact, these claims detract from the paper's empirical contributions). End instead on another claim more centrally grounded in this study's empirical data. For instance, here's one such claim: "based on simulations, girls scored in the top 10% of STEM grades as often as boys" (of course, as supported by Figure 3). Or the authors could simply note that findings will be discussed in terms of talent/ability/other literature on test scores (without having to make any specific claims about those points in the abstract).

(2) THE NEW TEST SCORE ANALYSIS RAISES MANY MORE QUESTIONS

The PISA test score analysis is an interesting addition, but its findings about variability in STEM versus non-STEM subjects contradict most prior research without an explanation for why.

PRIOR STUDIES: As the current authors themselves acknowledge, "evidence from standardized tests administered to children and adolescents indicates a greater gender difference in variation in performance in STEM subjects than other subjects" (lines 65-67; this manuscript). For instance, one of these prior studies (Hedges & Nowell, 1995; which the authors appropriately cite) found that, "males were found to be more variable in almost all domains, with slightly greater differences in variability in quantitative domains (5 to 25% more variable) than reading (3 to 16%) and nonverbal reasoning (4 to 15%)," as summarized by Lakin (2013, p. 264).

INTERNAL CONTRADICTIONS: These findings (i.e., greater sex difference in variability in STEM subjects) contrasts with this manuscript's findings both about grades and PISA test scores. In other words, the manuscript has an internal contradiction because it later states that, "girls' test scores were more consistent than boys, with the gender difference in variability being significantly greater in non-STEM than STEM subjects" (lines 238-240).

OTHER RECENT TEST SCORE ANALYSES: At first, I thought this discrepancy with prior research might be explained by historical differences (e.g., Hedges & Nowell was published almost 25 years ago). But then I found other recent large-scale nationally representative analyses of test scores finding similar results. For instance, consider Lakin's (2013) analysis of the Cognitive Abilities Test. In the most recent national sample in 2011 with included over 45,000 children (CogAT 7), Lakin found that that boys were 53% more variable than girls in quantitative reasoning (a huge difference!), but only 13% more variable in verbal reasoning. Similar differences in variability between quantitative vs. verbal reasoning were found in the 1992 and 2000 samples.

Lakin, J. M. (2013). Sex differences in reasoning abilities: Surprising evidence that male-female ratios in the tails of the quantitative reasoning distribution have increased. *Intelligence*, 41, 263-274.

UNCLEAR WHY THESE DISCREPANCIES EXIST: None of these findings necessarily imply the authors' analysis was done incorrectly. To the contrary, PISA is a large high-quality data source and the authors' analysis appears to be appropriate without any obvious technical flaws. Hence, I'm simply confused by these wildly divergent results. The PISA analysis raises so many more questions: why have prior test scores analyses found opposite results? Has the results for PISA changed over time? What about TIMSS (another large international dataset of test scores)? How do the test score results vary across nations? For instance, would the PISA analysis find results consistent with prior research if only the PISA U.S. sample was analyzed?

MY RECOMMENDATION: In my view, answering these questions is simply beyond the scope of this current work. Hence, I personally recommend removing the PISA analysis and instead restricting this study's empirical conclusions to only grades. I would still cite the relevant test score papers (e.g., Hedges & Nowell, 1995; Lakin, 2013) and simply say it's unclear why they find different results regarding variability in STEM versus non-STEM subjects. I agree with Reviewer #2 that focusing the empirical analyses on only grades is sufficient, given the relative dearth of prior research on grades compared to test scores.

(3) SENSITIVITY OF RESULTS TO INCLUSION CRITERIA

Having the data and analysis scripts available greatly helped me evaluate the empirical robustness of the authors' claims and address my own earlier reservations (e.g., regarding nesting). However, one of the core empirical claims showed some sensitivity to inclusion criteria: in the full sample of

studies (including both pre-college and college samples), the sex difference in variability did not significantly differ between STEM versus non-STEM subjects.

More specifically, I'm referring to lines 2383-2401 in MA_STEP3_models.rmd. Running that code finds no significant difference in lnCVR between STEM and non-STEM subjects ($p = .229$) in the full sample of studies. In fact, in non-STEM university courses, girls' grades were *more* variable than boys, though not significantly so (Table S10 in the SI; also Figure S11, Part E). In other words, one of the core empirical results depends on excluding the university samples.

However, I'm not too concerned about this sensitivity because this difference in results seems to be largely driven by one outlier university study ($study_ID = s049$). That study could also explain the very large CI for language subjects at the university level in Figure S11E. Once that study was removed from analysis, the difference in lnCVR between STEM and non-STEM subjects reappeared ($\text{diff in lnCVR} = 0.076$; $p < .0001$) when analyzing the full sample of studies (minus the outlier).

MY RECOMMENDATION: I would briefly note this empirical sensitivity using one or two sentences in the main text and then elaborate on this point in the SI. I understand the rationale for focusing on the school (non-university) data in the moderator analyses (lines 137-141), but readers such as myself may likely wonder if the empirical results hinge on that analytic decision.

(3) USE OF THE TERM "BIAS":

The authors use the term "bias" throughout the manuscript to apparently refer to "overrepresentation" (e.g., "STEM is male-biased because a greater proportion of males than females"; "male bias amongst the top-achieving students"). I would avoid using the term "bias" in this way because "gender bias" has usually meant gender discrimination (e.g., evaluating grant applications differently based on author sex). Hence, I would avoid using the term "bias," except in cases where the authors are referencing discrimination (which appears to be generally not the case in this manuscript).

(4) OTHER COMMENTS

The following comments are more minor concerns intended to help further improve the manuscript, but are not essential to my recommendation to publish this manuscript or not.

- Line 41: The term "occupational segregation" applies to actual differences in the workforce, but not differences in career aspirations (which is what the previous sentence refers to).
- Lines 87-88: The use of causal language (e.g., grades "have a greater impact on students' academic self-concept") is not appropriate given the correlational nature of the data.
- Lines 190-199: Most of the findings regarding year and age were non-significant. I would phrase them even more tentatively (e.g., rather than using language like "marginally decreased"). Same point goes for line 185 ("weakly affected by the year of study"...still implies there was a reliable, though weak, effect).
- Lines 250-252: That's a false dichotomy (i.e., gender gaps in expectations of success could arise from both backlash effects, ability tilt, and other variables rather than only backlash or ability tilt).

Reviewer #2 (Remarks to the Author):

I have previously reviewed this manuscript, so I won't repeat my discussion about the importance and novelty of this research.

This is a *substantial* improvement, and the author(s) have addressed all of my concerns that I raised. I have also read the response to the other reviewers (particularly reviewer 1) and the subsequent analysis of PISA test data. This really has gone above and beyond what would

normally be expected, and addressed my concerns about the discrepancy between grade and test scores. The PISA analysis presented in supplementary could well have been a unique publication output in itself.

I recommend that this article be accepted, and look forward to citing it one day in Nature Communications.

Reviewer #3 (Remarks to the Author):

I have read the manuscript and review letter thoroughly. All points appear to be appropriately addressed. I recommend publication.

**Reviewer 1, Comment 1**

*However, two major concerns remain: (1) the claims about talent and ability, and*
*(2) the new PISA test score analysis raises more questions than it answers. The*
*manuscript has been improved in the first regard, but it still makes claims that go*
*beyond the scope of this project, especially in the abstract (which is what most*
*readers will see). The second concern is new because the PISA test score analysis is*
*new. If these two major concerns are addressed, I would recommend publication,*
*but I cannot do so in the current state.*

**Response to Reviewer 1, Comment 1**

We have addressed both of these concerns. (1) We have changed wording in the
abstract and main text (see our response to Comment 2). (2) We have closely
examined the PISA tests scores, and improved the analyses, although at the editor's
request we have not removed this analysis from this revised manuscript. We discuss
this issue extensively in our response to Comment 3.

**Reviewer 1, Comment 2**

*(1) CLAIMS ABOUT TALENT AND ABILITY*

*The manuscript has been improved now that it explicitly acknowledges the test-*
*grade discrepancy and potential reasons for it. The claims about talent and ability*
*are now far more nuanced and appropriate given the complexity of the broader*
*data. However, there are places still where the claims go beyond the data by*
*inappropriately conflating grades with talent and ability.*

*THE ABSTRACT. My biggest concern is the last sentence of the abstract: "Given*
*the similar distribution of STEM grades for girls and boys, the gender bias in those*

*employed in standard careers in STEM is unlikely to be driven by fewer girls than*
*boys showing the requisite talent.” The manuscript later adds nuance to this claim*
*by noting how student behavior can influence grades (e.g., lines 84-85). However,*
*the abstract must be evaluated as a critical standalone paragraph, and in this case*
*the claims go beyond the current focus and scope of this project.*

*MANUSCRIPT: In addition, there are places in the manuscript where grades are*
*referred to “academically measured ability” (lines 217-220) and “academic ability”*
*(lines 273-275). I applaud the authors for emphasizing how grades might shape*
*self-concepts and predict later outcomes such as graduating college (those are*
*compelling points). But still I’d argue referring to grades as “ability” is not*
*appropriate.*

*My concerns echo Reviewer #3’s earlier comment: “grades have also been heavily*
*scrutinized by due to their overwhelming subjectivity. Multiple factors that are*
*independent of actual ability influence the grades students obtain. Student*
*behavior, for example, influences teacher assignment of grades. Girls tend to be*
*less disruptive, more organized, and more studious.”*

*MY RECOMMENDATION: My recommendation is simple: avoid making claims*
*about talent and ability in the abstract. The study is sufficiently noteworthy without*
*having to make such claims (and, in fact, these claims detract from the paper’s*
*empirical contributions). End instead on another claim more centrally grounded in*
*this study’s empirical data. For instance, here’s one such claim: “based on*
*simulations, girls scored in the top 10% of STEM grades as often as boys” (of*
*course, as supported by Figure 3). Or the authors could simply note that findings*
*will be discussed in terms of talent/ability/other literature on test scores (without*
*having to make any specific claims about those points in the abstract).*

**Response to Reviewer 1, Comment 2**

ABSTRACT: We thank the reviewer for these suggestions, and we have replaced the
final sentence of the Abstract. It now reads: "*Simulations of these differences suggest
the top 10% of a class contains equal numbers of girls and boys in STEM, but more
girls in non-STEM subjects. Gender differences in the distributions of grades could
have implications for students' subsequent investment in STEM.*" (Lines 20-24)

MANUSCRIPT: We have replaced mentions of 'ability' in the manuscript with
'achievement' or 'grades'. We have also added in the discussion that academic
achievement is likely to be an imperfect measure of the ability to actually work in a
given field: "*Therefore, while our data does not preclude a gender gap amongst the
exceptionally talented, it nevertheless indicates a practical similarity in girls' and
boys' academic achievements, which are likely to provide an imperfect but valid
measure of the ability to pursue STEM*" (lines 221-224). This seems a fair
compromise, because we do not think many people in academia would argue that
grades and tests scores provide no information about ability.

**Reviewer 1, Comment 3**

(2) *THE NEW TEST SCORE ANALYSIS RAISES MANY MORE QUESTIONS*

*The PISA test score analysis is an interesting addition, but its findings about*
*variability in STEM versus non-STEM subjects contradict most prior research*
*without an explanation for why.*

*PRIOR STUDIES: As the current authors themselves acknowledge, “evidence from*
*stardardised tests administered to children and adolescents indicates a greater*
*gender difference in variation in performance in STEM subjects than other*
*subjects” (lines 65-67; this manuscript). For instance, one of these prior studies*
*(Hedges & Nowell, 1995; which the authors appropriately cite) found that, “males*
*were found to be more variable in almost all domains, with slightly greater*
*differences in variability in quantitative domains (5 to 25% more variable) than*
*reading (3 to 16%) and nonverbal reasoning (4 to 15%),” as summarized by Lakin*
*(2013, p. 264).*

*INTERNAL CONTRADICTIONS: These findings (i.e., greater sex difference in*
*variability in STEM subjects) contrasts with this manuscript’s findings both about*
*grades and PISA test scores. In other words, the manuscript has an internal*
*contradiction because it later states that, “girls’ test scores were more consistent*
*than boys, with the gender difference in variability being significantly greater in*
*non-STEM than STEM subjects” (lines 238-240).*

*OTHER RECENT TEST SCORE ANALYSES: At first, I thought this discrepancy*
*with prior research might be explained by historical differences (e.g., Hedges &*
*Nowell was published almost 25 years ago). But then I found other recent large-*
*scale nationally representative analyses of test scores finding similar results. For*
*instance, consider Lakin’s (2013) analysis of the Cognitive Abilities Test. In the*
*most recent national sample in 2011 with included over 45,000 children (CogAT 7),*
*Lakin found that that boys were 53% more variable than girls in quantitative*

*reasoning (a huge difference!), but only 13% more variable in verbal reasoning.*
*Similar differences in variability between quantitative vs. verbal reasoning were*
*found in the 1992 and 2000 samples.*
*Lakin, J. M. (2013). Sex differences in reasoning abilities: Surprising evidence that*
*male-female ratios in the tails of the quantitative reasoning distribution have*
*increased. Intelligence, 41, 263-274.*
*UNCLEAR WHY THESE DISCREPANCIES EXIST: None of these findings*
*necessarily imply the authors' analysis was done incorrectly. To the contrary, PISA*
*is a large high-quality data source and the authors' analysis appears to be*
*appropriate without any obvious technical flaws. Hence, I'm simply confused by*
*these wildly divergent results. The PISA analysis raises so many more questions:*
*why have prior test scores analyses found opposite results? Has the results for PISA*
*changed over time? What about TIMSS (another large international dataset of test*
*scores)? How do the test score results vary across nations? For instance, would the*
*PISA analysis find results consistent with prior research if only the PISA U.S.*
*sample was analyzed?*
*MY RECOMMENDATION: In my view, answering these questions is simply*
*beyond the scope of this current work. Hence, I personally recommend removing*
*the PISA analysis and instead restricting this study's empirical conclusions to only*
*grades. I would still cite the relevant test score papers (e.g., Hedges & Nowell, 1995;*
*Lakin, 2013) and simply say it's unclear why they find different results regarding*
*variability in STEM versus non-STEM subjects. I agree with Reviewer #2 that*
*focusing the empirical analyses on only grades is sufficient, given the relative*
*dearth of prior research on grades compared to test scores.*

**Response to Reviewer 1, Comment 3**

We have taken the reviewer’s concerns seriously, and we closely compared our PISA
 analysis to the methods and results presented in Hedges (Hedges & Nowell 1995) and
 Lakin (Lakin 2013). Both Hedges and Lakin used the variability ratio – $VR = \frac{s_m^2}{s_f^2}$ – to
 test for gender differences in variance. As the reviewer points out, both these analyses
 found very large gender differences. In Figure 1 below, we have plotted the raw
 values of variability ratios from both papers.

**Figure 1**

*Distribution of variability ratios (VR) expressed as the percentage of greater male*
 *variability for three subject types: “other” (grey), “non-STEM” (purple), and*
 *“STEM” (green). (A) Data from Table 1, p.265 of Lakin 2013. We have classified*
 *subjects as: Other (n = 4) = Nonverbal; non-STEM (n = 4) = Verbal; STEM (n = 4)*
 *= Quantitative. (B) Data from Table 2, p.43 of Hedges 1995. We have classified*
 *subjects as: Other (n = 19) = Associative memory, Auto and shop information,*
 *Electronics information, Mechanical reasoning, Nonverbal reasoning, Perceptual*
 *speed, Spatial ability; non-STEM (n = 11) = Reading comprehension, Social studies,*
 *Vocabulary; STEM (n = 10) = Mathematics and Science.*

Both panels of Figure 1 show larger variability differences for STEM subjects, but the
scale of the y-axis makes this difficult to see because of some very large values from
less conventional test scores in Hedges' data, where students might be tested on
material that is not taught in the classroom. For the remaining discussion of these
data, we focus on the subject classifications most closely resembling those we used in
our meta-analysis of school grades (Figure 2).

**Figure 2**

*Distribution of variability ratios (VR) expressed as the percentage of greater male*
*variability for two subject types: "non-STEM" (purple), and "STEM" (green).*

*Details for panels (A) and (B) are the same as in Figure 1, with the subject type*
*"other" removed*

The difference between these results and our results lies in the **magnitude** of the
overall differences, and the **direction** of the STEM-non-STEM difference. It is

possible that a mean-variance relationship could explain both of these differences, at

least to some degree.

**Mean-variance relationship**

We analysed our grades data with *lnCVR*, rather than *VR*, because there was a strong
positive correlation between means and variances (Supplementary Figures 1 & 2 in
the supplementary materials: lines 247-261). We cannot directly check for this
correlation in Hedges' and Lakin's data, because their tables do not present means
and variances, but we can check for a relationship between mean differences (Cohen's
*d*) and variance differences (*VR*):

**Figure 3**

*The relationship between mean differences (*d*) and variance differences (*VR*) for non-*
*STEM (purple squares) and STEM (green circles) subjects. Positive values of *d*, and*
**VR* values greater than 1, indicate higher means and greater variance for boys. The*
*dashed grey line shows the linear relationships for all the data combined, whereas the*
*solid lines show the relationship within each subject category. The axis margins are*
*the same in both panels. Information about (A) Lakin's data and (B) Hedges' data is*
*the same as above (Figure 1).*

From Figure 3 above, we can see that overall there is a positive relationship between
mean and variance differences, which seems to be largely driven by STEM subjects.

This suggests there is a mean-variance relationship in the raw data. We can also see
the expected pattern for mean differences in STEM subjects for test scores: boys tend
to do better on STEM tests (more green points to the right of $d = 0$), whereas in non-
STEM tests there is either no mean difference, or an advantage for girls (more purple
squares are left of $d = 0$).

We therefore conclude that Lakin's and Hedges' analysis might have such large
estimates for VR - particularly for STEM - because of the combined effects of mean
and variance differences; boys are both inherently more variable, but they also receive
higher mean test scores, which further inflates a gender difference in variability. Were
these data to be analysed with *lnCVR*, we predict the variability differences will be
more modest, and the difference between STEM and non-STEM might even
disappear. Of course, this is speculation, and we agree with the reviewer that a proper
discussion and analysis of these differences is material for a separate paper. Please
note that our choice of the PISA dataset was pragmatic: we could not readily obtain
sample sizes, means, and standard deviations for other large-scale datasets of test
scores (including the dataset – TIMSS – mentioned by the reviewer).

**PISA Data**

Having explored Hedges' and Lakin's data, we were surprised to find no positive
mean-variance relationship in our PISA dataset. We present the comparable plot to
Figure 3 in Figure 4.

**Figure 4**

*The relationship between mean differences (d) and variance differences (VR) for non-*
 *STEM (purple squares) and STEM (green circles) subjects in the 2015 PISA dataset.*
 *Positive values of d , and VR values greater than 1, indicate higher means and greater*
 *variance for boys. The dashed grey line shows the linear relationships for all the data*
 *combined, whereas the solid lines show the relationship within each subject category.*

A model of the raw values for means and variance confirmed the absence of a strong
 mean-variance relationship. It is quite unusual not to see a positive mean-variance
 relationship in this type of data, so we contacted PISA through their website to ensure
 we were extracting the correct data. Through this correspondence we found an easier
 way to obtain sample sizes, so the data in the revised manuscript is slightly different,
 but not noticeably so.

The absence of a positive correlation between mean and variance makes results from
 an analysis of $\ln CVR$ hard to interpret. **Therefore, in the revised manuscript we**
 **have corrected the PISA analysis to test for variability differences using the**
 **more appropriate log variability ratio, $\ln VR$.** Analyses with $\ln CVR$ are available in
 the updated online repository:

https://osf.io/vu8h2/?view_only=b5131c883bd24b90919c4ea4944e93c5. The most
noticeable change when using *lnVR* is that the magnitude of greater male variability is
no longer different between non-STEM and STEM. Also, please note that **the use of**
***lnVR* makes it easier for the reader to compare the result of the PISA dataset**
**with those of earlier work** (e.g., Lakin’s and Hedges’ analysis). Having said this, we
are willing to reinstate the original *lnCVR* analyses, or we could prepare a revised
version of the manuscript with all PISA analyses removed.

After investigating this topic further, we now think these PISA analyses are better
suited to a separate paper where we could give this topic greater attention, rather than
relegating it to the supplementary materials where it will be seen by few readers (i.e. in
agreement with Reviewer 2, who wrote “*The PISA analysis presented in*
*supplementary could well have been a unique publication output in itself.*”) As the
PISA analysis was not part of our original manuscript, and the general consensus of
the reviewers is that one of our manuscript’s key strengths is its focus on grades
(rather than test scores like PISA), we think the supplementary materials are less
coherent with the substantial addition of the PISA analysis. As such, our preference is
to remove the PISA analysis and publish it elsewhere, but of course we are happy to
leave the analysis in, if required for publication in *Nature Communications*. Below we
show the updated version of the PISA section of the SI:

SI, lines 195-245:

**Analysis of test scores – 2015 PISA**

To explore whether gender differences in variability across STEM and non-STEM are
broadly applicable to school achievement, and not confined to school grades which

tend to favour girls, we analysed data from the 2015 Programme for International
 Student Assessment (referred to as PISA hereafter). PISA is an international
 measurement of achievement on standardised tests by 15-year-old students.

We downloaded results tables for test performance in reading, mathematics, and
 science subjects, from the PISA 2015 Results (Volume I) (OECD 2016), and
 extracted the means and standard deviation in achievement for boys and girls in each
 jurisdiction (country). We obtained the corresponding sample sizes from separate
 tables, available within the “Questionnaire items” from the Compendia on the PISA
 2015 Database webpage (OECD 2015). To calculate sample sizes we used the number
 of “valid” students for each subject type, and the given percentage of boys and girls
 within each jurisdiction.

Overall, the PISA dataset summarised test scores in maths, reading and science for
 226,131 female and 226,480 male students. The students were tested from 64
 jurisdictions. The minimum and maximum number of students in a jurisdiction was
 3,371 and 23,141, respectively.

To test for gender differences in means we used the same metric as our main meta-
 analysis ($\ln RR$). To test for differences in variance we used a different metric, because
 there was no consistent mean-variance relationship in the PISA dataset (in contrast to
 our main dataset). We therefore used the log variability ratio ($\ln VR$) and its sampling
 variance ($s^2_{\ln VR}$) to test for gender differences in variance (Nakagawa et al. 2015),
 where:

$$\ln VR = \ln \left(\frac{s_f}{s_m} \right) + \frac{1}{2(n_f - 1)} - \frac{1}{2(n_m - 1)}$$

$$s_{\ln VR}^2 = \frac{1}{2(n_f - 1)} + \frac{1}{2(n_m - 1)}$$

281 We also directly modelled between-subject differences in variability by testing for the
 282 moderating effects of sex and subject on the logged standard deviation ($\ln SD$) in test
 283 scores (Raudenbush & Bryk 1987):

$$\ln SD = \ln s + \frac{1}{2(n - 1)}$$

$$s_{\ln SD}^2 = \frac{1}{2(n - 1)}$$

We fitted meta-analytic models to each effect size with the same approach as our
 main analysis (using the *rma.mv* and *robust* functions from the *metafor* package (v.
 2.0.0) in *R* (v. 3.4.3) (Viechtbauer 2010)). We accounted for non-independence
 arising from multiple effect sizes from the same study by fitting the ID of the
 jurisdiction and a comparison ID as random effects. We modelled sampling variances
 with a covariance matrix, assuming a 0.5 correlation between variances arising from
 the same jurisdiction. To test for differences between STEM and non-STEM subjects,
 we fitted univariate meta-regression models by including subject as a fixed effect,
 where subject was either STEM (maths and science) or non-STEM (reading).

**PISA Results**

Full results are shown in Supplementary Table 15. Overall, girls sitting the 2015 PISA
 received 2% higher scores than boys ($\ln RR_{\text{overall}(\text{mean})}$ CI: 1.4% to 2.7%; Supplementary
 Fig. 12A), with 6.4% less variation among girls than among boys ($\ln VR_{\text{overall}(\text{variance})}$): CI:
 5.6% to 7.2%; Supplementary Fig. 12B).

Girls' small advantage in PISA tests scores was entirely driven by a 6.9% advantage
in non-STEM ($\ln RR_{STEM(\text{mean})}$ CI: 6% to 7.7%). In contrast, girls' showed a non-
significant 0.3% *disadvantage* in STEM ($\ln RR_{\text{non-STEM}(\text{mean})}$ CI: -0.9% to 0.2%;
Supplementary Fig. 12A).

While girls' test scores were significantly more consistent across subjects, this
variability gap was no different between non-STEM and STEM subjects ($\ln VR_{\text{non-STEM}}$
$_{STEM \text{ diff}}$ CI: -0.7% to 0.3%). When looking at the variability for girls and boys
separately, both girls' and boys' scores were more variable in non-STEM than STEM
subjects, but the difference was smaller for girls (Supplementary Fig. 12C,
Supplementary Table 15).

Together these results suggest that the shape of the achievement distributions was
consistent, but in STEM subjects boys' test scores shifted to the right.

SI Figures, lines 331-345:

**Supplementary Figure 12**

Results of analyses on (A) ratios of the grade means, (B) ratios of grade variabilities,
 and (C) coefficients of variations for girls (red) and boys (blue) from the 2015 PISA,
 corresponding to Supplementary Table 15. Diamonds and circles represent meta-
 analytic estimates of mean effect sizes, and their 95% confidence intervals are drawn
 as whiskers. In panel A, natural logarithm of response ratio ($\ln RR$) represents the

average difference between girls’ and boys’ test scores; positive values of $lnRR$
 indicate lower boys’ tests scores. In panel **B**, natural logarithm of variation ratio
 ($lnVR$) represents the average difference in test score variation between boys and
 girls; negative values of $lnVR$ indicate greater male variance. In panel **C**, natural
 logarithms of standard deviation in test scores ($lnSD$) are shown for girls and boys to
 illustrate grade variation by gender; more negative values of $lnSD$ indicate less
 variation.

_____
 _____

SI Tables, lines 466-475:

**Supplementary Table 15**

Estimated effect sizes and heterogeneity estimates for meta-analytic (intercept-only)
 models, and meta-regression models with subject (STEM or Non-STEM) as
 moderators, for PISA test scores. $lnRR$ is a measure of mean difference between test
 scores for girls and boys. $lnVR$ is a measure of the difference in variability between
 girls and boys. $lnSD$ is the total variability for each Jurisdiction. I^2_{Total} represents
 proportion of variance not attributed to standard error. $I^2_{Jurisdiction}$ represents proportion
 of variance attributed to the Jurisdiction where students were tested. $I^2_{comp_ID}$
 represents residuals against sampling error.

Measure	Data	Fixed effects			Random effects			Heterogeneity		
		Mean	CI.lb	CI.ub		Sigma ²	N levels	I^2_{Total}	$I^2_{Jurisdiction}$	$I^2_{comp_ID}$
$lnRR$	Intercept	0.020	0.014	0.026	Jurisdiction	0.000	64	99.4	1.7	97.7
	Subject: STEM	0.066	0.058	0.075	comp_ID	0.002	192			

	Subject: Non- STEM	-0.003	-0.009	0.002						
	non- STEM - STEM difference	-0.070	-0.074	-0.066						
lnVR	Intercept	-0.066	-0.075	-0.057	Jurisdiction	0.001	64	79.3	75.3	4.0
	Subject: STEM	-0.065	-0.075	-0.054	comp_ID	0	192			
	Subject: Non- STEM	-0.067	-0.076	-0.058						
	non-STEM - STEM difference	-0.002	-0.008	0.003						
lnSD - overall	non- STEM - STEM difference	4.526	4.499	4.554	Jurisdiction	0.009	64	98.9	73.9	25.0
	girls-boys difference	0.068	0.057	0.078	comp_ID	0.003	192			
lnSD - girls	non- STEM - STEM difference	-0.047	-0.062	-0.032						
lnSD - boys	non- STEM - STEM difference	-0.044	-0.058	-0.030						

The main text of the manuscript only mentioned the PISA supplementary analysis in
one paragraph of the discussion. In the revised manuscript, we have made minor
changes to this paragraph (**highlighted**), to reflect the new results using *lnVR* rather
than *lnCVR*:

Discussion, lines 235-251:

We analysed school grades, where girls show a well-established advantage over boys
(Duckworth & Seligman 2006), whereas most previous tests of gender differences in
variability have focussed on test scores (Baye & Monseur 2016; Hedges & Nowell
1995; Reilly et al. 2015). To explore whether the smaller variability difference in
STEM compared to non-STEM is confined to school grades, we performed a
supplementary analysis of a large international dataset of standardised test scores of
15-year-olds (see Supplementary Note 2 for details). This supplementary analysis
**found gender differences in variance that were consistent across subjects**; girls' test
scores were more consistent than boys, **with equivalent gender differences in non-**
**STEM and STEM subjects** (Supplementary Fig. 12). However, girls only showed a
mean advantage in non-STEM. Therefore, it appears that the mean differences
between test scores and grades are caused by shifts in the position of girls' and boys'
distributions, **rather than changes in the shape of distributions in STEM compared to**
**non-STEM (girls' distributions of both grades and test-scores are narrower than boys'**
**distributions, but the difference is not more pronounced in STEM)**. If girls perceive
they have fewer competitors in non-STEM subjects because, on average, fewer boys
perform better than girls, this might lead to a preference for non-STEM over STEM
careers (Niederle & Vesterlund 2010; Gneezy & Rustichini 2004).

**Reviewer 1, Comment 4**

**(3) SENSITIVITY OF RESULTS TO INCLUSION CRITERIA**

*Having the data and analysis scripts available greatly helped me evaluate the*
*empirical robustness of the authors' claims and address my own earlier*
*reservations (e.g., regarding nesting). However, one of the core empirical claims*
*showed some sensitivity to inclusion criteria: in the full sample of studies (including*
*both pre-college and college samples), the sex difference in variability did not*
*significantly differ between STEM versus non-STEM subjects.*
*More specifically, I'm referring to lines 2383-2401 in MA_STEP3_models.rmd.*
*Running that code finds no significant difference in lnCVR between STEM and*
*non-STEM subjects ($p = .229$) in the full sample of studies. In fact, in non-STEM*
*university courses, girls' grades were **more** variable than boys, though not*
*significantly so (Table S10 in the SI; also Figure S11, Part E). In other words, one*
*of the core empirical results depends on excluding the university samples.*
*However, I'm not too concerned about this sensitivity because this difference in*
*results seems to be largely driven by one outlier university study (study_ID = s049).*
*That study could also explain the very large CI for language subjects at the*
*university level in Figure S11E. Once that study was removed from analysis, the*
*difference in lnCVR between STEM and non-STEM subjects reappeared (diff in*
*lnCVR = 0.076; $p < .0001$) when analyzing the full sample of studies (minus the*
*outlier).*
*MY RECOMMENDATION: I would briefly note this empirical sensitivity using*
*one or two sentences in the main text and then elaborate on this point in the SI. I*
*understand the rationale for focusing on the school (non-university) data in the*
*moderator analyses (lines 137-141), but readers such as myself may likely wonder if*
*the empirical results hinge on that analytic decision.*

**Response to Reviewer 1, Comment 4**

We thank the reviewer for this great pick-up in the supplementary analysis, and
concur with the reviewer’s conclusion about the influence of that single study. We
have followed the reviewer’s suggestions and changed the main text and SI Results
section accordingly:

Main text, lines 143-147:

“The results from analyses for the whole dataset, and only the university subset, are
provided in the SI (the university subset also had small sample sizes for STEM and
non-STEM subjects, making results from moderator analyses sensitive to outlier
studies).”

SI, lines 80-87:

“Additionally, data from the university was predominantly for “Global” grades; there
were only 13 data points for science, 10 for maths and 4 for language. These low
sample sizes resulted in broad confidence intervals for moderator analyses on the
effect of subject (e.g., Supplementary Fig. 11E), and sensitivity to outliers. In
particular, one influential study for language grade variability in the university subset
strongly influenced the difference in variability between STEM and non-STEM
subjects, causing there to be no significant differences in variability in non-STEM
(Supplementary Table 10).”

**Reviewer 1, Comment 5**

(3) *USE OF THE TERM “BIAS”:*

*The authors use the term “bias” throughout the manuscript to apparently refer to*
*“overrepresentation” (e.g., “STEM is male-biased because a greater proportion of*
*males than females”;* *“male bias amongst the top-achieving students”). I would*
*avoid using the term “bias” in this way because “gender bias” has usually meant*
*gender discrimination (e.g., evaluating grant applications differently based on*
*author sex). Hence, I would avoid using the term “bias,” except in cases where the*
*authors are referencing discrimination (which appears is generally not the case in*
*this manuscript)*

**Response to Reviewer 1, Comment 5**

In all cases where there is potential ambiguity we have replaced ‘bias’ with ‘skew’ or
‘overrepresented’.

**Reviewer 1, Comment 6**

*The following comments are more minor concerns intended to help further improve*
*the manuscript, but are not essential to my recommendation to publish this*
*manuscript or not.*

• *Line 41: The term “occupational segregation” applies to actual differences in the*
*workforce, but not differences in career aspirations (which is what the previous*
*sentence refers to).*

• *Lines 87-88: The use of causal language (e.g., grades “have a greater impact on*
*students’ academic self-concept”) is not appropriate given the correlational nature*
*of the data.*

• *Lines 190-199: Most of the findings regarding year and age were non-significant.*
*I would phrase them even more tentatively (e.g., rather than using language like*

*“marginally decreased”). Same point goes for line 185 (“weakly affected by the year*
*of study”...still implies there was a reliable, though weak, effect).*

• *Lines 250-252: That’s a false dichotomy (i.e., gender gaps in expectations of*
*success could arise from both backlash effects, ability tilt, and other variables*
*rather than only backlash or ability tilt).*

**Response to Reviewer 1, Comment 6**

Changed Line 41 (now Line 43): *“This phenomenon contributes to ‘occupational*
*segregation’, and there are numerous incentives to reduce its prevalence”*

Changed Lines 87-88 (now lines 87-91): *“teacher-assigned grades are likely to affect*
*students’ lives, and it is a reasonable conjecture that they have a greater impact on*
*students’ academic self-concept than standardised test scores (Möller et al. 2009).”*

We agree that we are citing observational data, and not data from an experiment
where students are assigned different grades and their future self-concept is then
examined. However, we think that most readers would accept that receiving low
grades does have an effect on how you assess your academic ability, and that students
receive these grades more often than scores from standardised tests.

Changed Lines 190-199 (now 196-197): We have changed *marginally* to *slightly*.

However, we have not changed Line 185 (now line 192) because there was a
significant difference between slopes for boys and girls [Lines 192-195: *“Within*
*genders, variability in grades showed a non-significant trend towards decreasing*

over time, but significantly more so for girls than boys (Supplementary Table 6,
Supplementary Figure 9G: $\ln CV_{study\ year\ boys-girls\ (slope\ diff)}: 0.032, CI: 0.004\ to\ 0.060$.”]

Changed Line 250 (now 254): we have replaced ‘or’ with ‘and/or’

In addition to these minor changes above, we noticed some inconsistencies in our
model specifications. We should have placed our correlation (covariance) matrix to
the level of sampling variance rather than to the level of residuals (within-effect-size
variance), but this was not done for all the models. Therefore, this specification has
been corrected in the updated manuscript. We only observed tiny quantitative changes
to our results (i.e. no qualitative changes).

**References**

Baye, A. & Monseur, C., 2016. Gender differences in variability and extreme scores
in an international context. *Large-scale Assessments in Education*, 4(1), p.541.

Duckworth, A.L. & Seligman, M.E.P., 2006. Self-discipline gives girls the edge:
Gender in self-discipline, grades, and achievement test scores. *Journal of*
*Educational Psychology*, 98(1), pp.198–208.

Gneezy, U. & Rustichini, A., 2004. Gender and competition at a young age. *American*
*Economic Review*, 94(2), pp.377–381.

Hedges, L.V. & Nowell, A., 1995. Sex differences in mental test scores, variability,
and numbers of high-scoring individuals. *Science*, 269(5220), pp.41–45.

Lakin, J.M., 2013. Sex differences in reasoning abilities: Surprising evidence that

male–female ratios in the tails of the quantitative reasoning distribution have
increased. *Intelligence*, 41(4), pp.263–274.

Möller, J. et al., 2009. A meta-analytic path analysis of the internal/ external frame of
reference model of academic achievement and academic self-concept. *Review of*
*Educational Research*, 79(3), pp.1129–1167.

Nakagawa, S. et al., 2015. Meta-analysis of variation: Ecological and evolutionary
applications and beyond R. B. O'Hara, ed. *Methods in Ecology and Evolution*,
6(2), pp.143–152.

Niederle, M. & Vesterlund, L., 2010. Explaining the gender gap in math test scores:
The role of competition. *Journal of Economic Perspectives*, 24(2), pp.129–144.

OECD, 2015. PISA 2015 Database. OECD Skills Surveys,
<http://www.oecd.org/pisa/data/2015database/>, accessed on 14th June 2018

OECD, 2016. PISA 2015 Results (Volume I): Excellence and Equity in Education.
Available at: <http://pisadataexplorer.oecd.org/ide/idepisa/>.

Raudenbush, S.M. & Bryk, A.S., 1987. Examining correlates of diversity. *Journal of*
*Educational Statistics*, 12(3), pp.241–269.

Reilly, D., Neumann, D.L. & Andrews, G., 2015. Sex differences in mathematics and
science achievement: A meta-analysis of National Assessment of Educational
Progress assessments. *Journal of Educational Psychology*, 107(3), pp.645–662.

Viechtbauer, W., 2010. Conducting meta-analyses in R with the metafor package.
*Journal of Statistical Software*, 36(3), pp.1–48.

REVIEWERS' COMMENTS:

Reviewer #1 (Remarks to the Author):

The authors continue to be highly responsive to the peer reviewer comments, which I greatly appreciate. All my major concerns have now been sufficiently addressed except for the PISA test score analyses. I completely agree with the authors that the PISA analyses are better suited for a separate paper, especially given the tight word limits for this journal.

More specifically, I agree with the authors that (a) adding those analyses makes this manuscript less coherent and (b) relegating them to the supplemental materials means most readers will not see them. As Reviewer #2 also noted, "the PISA analysis presented in supplementary could well have been a unique publication output in itself."

In addition, the test score analyses would need to go through a much more rigorous peer review process than they have so far. If the test score analyses were still included, I would recommend that the manuscript would need to be sent out again to the full review panel with specific attention focused on those analyses. That would create significantly more burden for this journal's staff as well as the peer reviewers.

Hence, it is my strong recommendation that those analyses be removed. Otherwise, this manuscript makes a rigorous important contribution to the research field.

Reviewer #2 (Remarks to the Author):

I have previously reviewed earlier versions of this manuscript. Then, as I do now, I find it to be of exceptionally high quality and the authors have gone above and beyond in including the PISA analysis. As much of this extra work has been to answer concerns by Reviewer 1, its almost stretched into a separate paper in length with the material given in the Supplementary Information. Sadly most readers never read the complete supplementary information, and will be unaware of the additional lengths this author/authors have gone to address the reviewers concerns. I would recommend accepting the paper in its entirety, so that it can be moved into publication.

The editor approaches me with a specific question however, which was chiefly whether the paper should be split into a second study. While this would offer benefits for the author (two publication outputs) the two sections are very strongly linked and really do work quite well together. I also feel that splitting it into a second paper would introduce an unreasonable publication delay (further peer-review), and that the authors have gone to great lengths in attempting to satisfy Reviewer 1's requests - some which have been reasonable, but others served only to delay the paper.

At the core of this article we have a single research question, does gender differences in variability differ for STEM and non-STEM subjects. The authors have presented MULTIPLE lines of evidence - firstly grades (which I think is perhaps the more interesting line of investigation, because it just hasn't been addressed before in the literature sufficiently well-enough that the question was answered), and then with PISA achievement scores. There was less variation in grades and achievement for girls than boys. There was larger variability in STEM grades than for non-STEM, but this was not found with the PISA achievement (keeping in mind the limitations of PISA, its only measuring reading as non-stem, and not other subjects). These are interesting, thought provoking findings, and will fuel future lines of research. Additionally the gender gap is quite minimal until you reach the upper right tail of the ability distribution (the authors chose upper 10% to investigate, other researchers such as Wai et al choose the upper 5%), and I think the authors do

a good job of presenting some reflection on this 274-284. There's a good balance between presenting the findings, and not exaggerating the nature of gender differences.

David Reilly
Griffith University

Reviewer #3 (Remarks to the Author):

I have read through the author comments and edits and feel that the authors have adequately addressed reviewer concerns. I recommend publication.